# Znhit1 controls intestinal stem cell maintenance by regulating H2A.Z incorporation

Bing Zhao[1,2], Ying Chen[1,3], Ning Jiang[1], Li Yang[1], Shenfei Sun[1,3], Yan Zhang[2], Zengqi Wen[4], Lorraine Ray[2], Han Liu[1], Guoli Hou[3] & Xinhua Lin[1,2]

Lgr5+ stem cells are crucial to gut epithelium homeostasis; however, how these cells are maintained is not fully understood. Zinc finger HIT-type containing 1 (Znhit1) is an evolutionarily conserved subunit of the SRCAP chromosome remodeling complex. Currently, the function of Znhit1 in vivo and its working mechanism in the SRCAP complex are unknown. Here we show that deletion of Znhit1 in intestinal epithelium depletes Lgr5+ stem cells thus disrupts intestinal homeostasis postnatal establishment and maintenance. Mechanistically, Znhit1 incorporates histone variant H2A.Z into TSS region of genes involved in Lgr5+ stem cell fate determination, including *Lgr5*, *Tgfb1* and *Tgfbr2*, for subsequent transcriptional regulation. Importantly, Znhit1 promotes the interaction between H2A.Z and YL1 (H2A.Z chaperone) by controlling YL1 phosphorylation. These results demonstrate that Znhit1/H2A.Z is essential for Lgr5+ stem cell maintenance and intestinal homeostasis. Our findings identified a dominant role of Znhit1/H2A.Z in controlling mammalian organ development and tissue homeostasis in vivo.

[1] State Key Laboratory of Genetic Engineering, School of Life Sciences, Zhongshan Hospital, Fudan University, Shanghai 200438, China. [2] Division of Developmental Biology, Cincinnati Children's Hospital Medical Center, Cincinnati, OH 45229, USA. [3] State Key Laboratory of Membrane Biology, Institute of Zoology, Chinese Academy of Sciences, Beijing 100101, China. [4] National Laboratory of Biomacromolecules, CAS Center for Excellence in Institute of Biophysics, Chinese Academy of Sciences, Beijing 100101, China. These authors contributed equally: Bing Zhao, Ying Chen, Ning Jiang. Correspondence and requests for materials should be addressed to B.Z. (email: bingzhao@fudan.edu.cn) or to X.L. (email: xlin@fudan.edu.cn)

The adult small intestinal epithelium is composed of two compartments: differentiated villi and proliferating crypts. Continuous renewal of the adult intestinal tissue is supported by Lgr5+ intestinal stem cells (ISCs) positioned at the bottom of the crypts. These adult ISCs divide to generate stem cells and transit amplifying daughter cells, which further give rise to various terminally differentiated progenies: absorptive enterocyte, secretory goblet, enteroendocrine, tuft, and Paneth cells[1–6]. Several signaling pathways, including Wnt[7–10], PI3K-Akt[11,12], transforming growth factor beta (TGF-β)[13–15], and Notch[7,16,17], have been demonstrated to regulate the self-renewal and differentiation of Lgr5+ ISCs. However, it is still unclear how epigenetic regulators and chromatin remodeling factors determine the fate of these cells through altering gene expression patterns. In particular, the role played by histone variants in controlling Lgr5+ ISC signature gene expressions, thus fate determination remains to be completely unknown.

In addition to the importance of understanding how intestinal homeostasis is maintained in adults, it is equally important to investigate the origin and specification of intestinal epithelial cells, especially Lgr5+ ISCs. The development of intestinal epithelium is initiated by forming of flat luminal epithelium in primitive gut tube (E9.5–E14.5)[18,19]. Then, differentiation-derived remodeling takes place and gives rise to villi and inter-villi (E15.5–E18.5). It has been shown that Lgr5+ ISC progenitors appear at the inter-villi regions in a Wnt-dependent manner during this stage[20–23]. In the end, Lgr5+ ISCs are generated (E18.5–Postnatal D7), which finalize the establishment of crypts and adult epithelium homeostasis[3]. Among these developmental processes, the mechanism of how Lgr5+ ISCs are generated after birth is poorly understood.

Modifications of chromatin structures allowing transcriptional activation or repression of particular sets of genes are essential for stem cell self-renewal and differentiation[24,25]. At the molecular level, this structural change can be brought by the replacement of canonical histone H2A with histone variant H2A.Z, which leads to chromatin remodeling and subsequent gene expression changes[26–30]. It is currently unknown about the incorporation or function of H2A.Z in mammalian organogenesis and tissue homeostasis. Previous studies have shown that both Znhit1 and YL1 are components of SRCAP (SNF-2 related CBP activator protein) complex, which can regulate the incorporation of H2A.Z into chromosome[31–34]. Moreover, a previous study in cultured myoblasts has shown that Znhit1 can bind to *myogenin* promoter and mediate H2A.Z incorporation for its expression[35]. However, as the genetic loss of function mutant mouse for Znhit1 is currently unavailable, the in vivo role of Znhit1 in development and tissue homeostasis is completely unknown. Furthermore, it is also unclear about the mechanism(s) of how Znhit1 and YL1 act in SRCAP complex to influence H2A.Z incorporation.

In this study, we establish Znhit1 conditional knockout mouse strain and examine its role in intestinal epithelium homeostasis establishment and maintenance. We show that Znhit1 supports Lgr5+ ISCs through regulating the expression of *Lgr5*, *Tgfb1*, and *Tgfbr2*, which are critical genes involved in Lgr5+ ISC fate determination. We further demonstrate that Znhit1 mediates the incorporation of histone variant H2A.Z into the TSS regions of these genes for transcriptional regulation. Our findings establish the essential role of Znhit1/H2A.Z in controlling Lgr5+ ISC maintenance and intestinal homeostasis, which implicates a therapeutic target in the intervention of gastrointestinal epithelium-related diseases.

## Results

**Znhit1 deletion disrupts postnatal generation of Lgr5+ ISC.** To determine the expression pattern of Znhit1 in intestinal epithelium, we performed *Znhit1* in situ in 8-week-old C57BL/6 mouse intestine section and found that the Znhit1 transcription was greatly enriched at the bottom of crypts (Fig. 1a). Consistently, RT-qPCR revealed that *Znhit1* mRNA was abundant in isolated crypts compared to villi (Supplementary Fig. 1a). Then, we dissociated *Lgr5-EGFP-IRES-creERT2*[1] crypts into single cells and sorted Lgr5+ ISCs (GFP$^{hi}$), daughter progenitor cells (GFP$^{low}$), and other crypt cells (GFP$^{neg}$) using FACS (Supplementary Fig. 1b). We found that Lgr5+ ISCs had robust Znhit1 expression, while their daughter progenitor cells and other crypt cells had significantly reduced Znhit1 expression (Supplementary Fig. 1b). This ISC-enriched expression pattern suggests that Znhit1 might be involved in the regulation of Lgr5+ ISC fate determination.

To investigate the functions of Znhit1 in intestinal development and homeostasis, we generated Znhit1 conditional knockout mice by inserting two *loxp* sites into upstream of exon 3 and downstream of exon 5 (Supplementary Fig. 2a), then employed *Villin-cre* to generate gut epithelium-specific deletion. The knockout efficiency was confirmed through the examination of Znhit1 mRNA and protein levels (Supplementary Fig. 2b–d). *Znhit1*$^{fl/fl}$; *Villin-cre* mice were born normally but exhibited intestinal epithelium dysfunction after birth: dramatic body weight decrease leads to 30% mice death following the first postnatal week (Fig. 1b, c), while the survived ones showed obvious growth retardation at P30 (Fig. 1c). As shown in Fig. 1d, intestinal villi and inter-villi structures appear to be normal in both *Znhit1*$^{fl/+}$; *Villin-cre* and *Znhit1*$^{fl/fl}$; *Villin-cre* mice at E18.5, indicating that Znhit1 had no obvious effect on embryonic development of intestinal epithelium. However, during the postnatal crypt morphogenesis stage, enlarged crypts and defective villi were observed in *Znhit1*$^{fl/fl}$; *Villin-cre* mice at P9 (Fig. 1d, e). This failed establishment of postnatal intestinal epithelium homeostasis well explained the phenotype of individuals.

Defective villi could be due to either impaired terminal differentiation or crypt dysfunction[11]. We found that the terminal differentiation of enterocytes (pan-differentiation marked by Krt20), goblet cells (marked by Mucin2), or enteroendocrine cells (marked by Chr-A) was not affected by Znhit1 deletion (Fig. 1e and Supplementary Fig. 3). To further examine whether these Znhit1-deficient crypts are functional, we mechanically dissociated intestinal crypts from *Znhit1*$^{fl/+}$; *Villin-cre* and *Znhit1*$^{fl/fl}$; *Villin-cre* mice at P9 and subjected them to in vitro culture according to the previous report[2]. We found that newly-formed tiny crypts isolated from *Znhit1*$^{fl/+}$; *Villin-cre* mice could efficiently survive and give rise to intestinal organoids, while the enlarged crypts isolated from *Znhit1*$^{fl/fl}$; *Villin-cre* mice completely lost this ability (Fig. 1f).

Lgr5+ intestinal ISCs are critical for both crypt function maintenance in vivo and organoid construction in vitro[1,2]. Therefore, we hypothesized that Znhit1 deficiency disrupted the postnatal establishment of functional crypts through restricting Lgr5+ ISCs. To test this, we employed *Lgr5-EGFP-IRES-creERT2* strain[1] to reveal the Lgr5+ ISCs and found that Lgr5+ ISCs were depleted in Znhit1-deficient intestinal epithelium at P9 (Fig. 1g). In addition, quantitative in situ assay showed that deletion of Znhit1 led to diminished mRNA expression of *Lgr5* and *Olfm4* (a robust Lgr5+ ISC marker[36,37]) at the bottom of crypts (Fig. 1h). Interestingly, Znhit1 deficiency had no restricting effect on expression of *Lgr5*, *Ascl2*, or *Olfm4* at P0 but initiated *Lgr5* downregulation after birth (Fig. 1i, j), suggesting that Znhit1 is required for the postnatal generation of Lgr5+ ISCs rather than on embryonic development of Lgr5+ progenitor cells. These data suggest that Znhit1 is essential for Lgr5+ ISC postnatal generation thus functional crypts establishment.

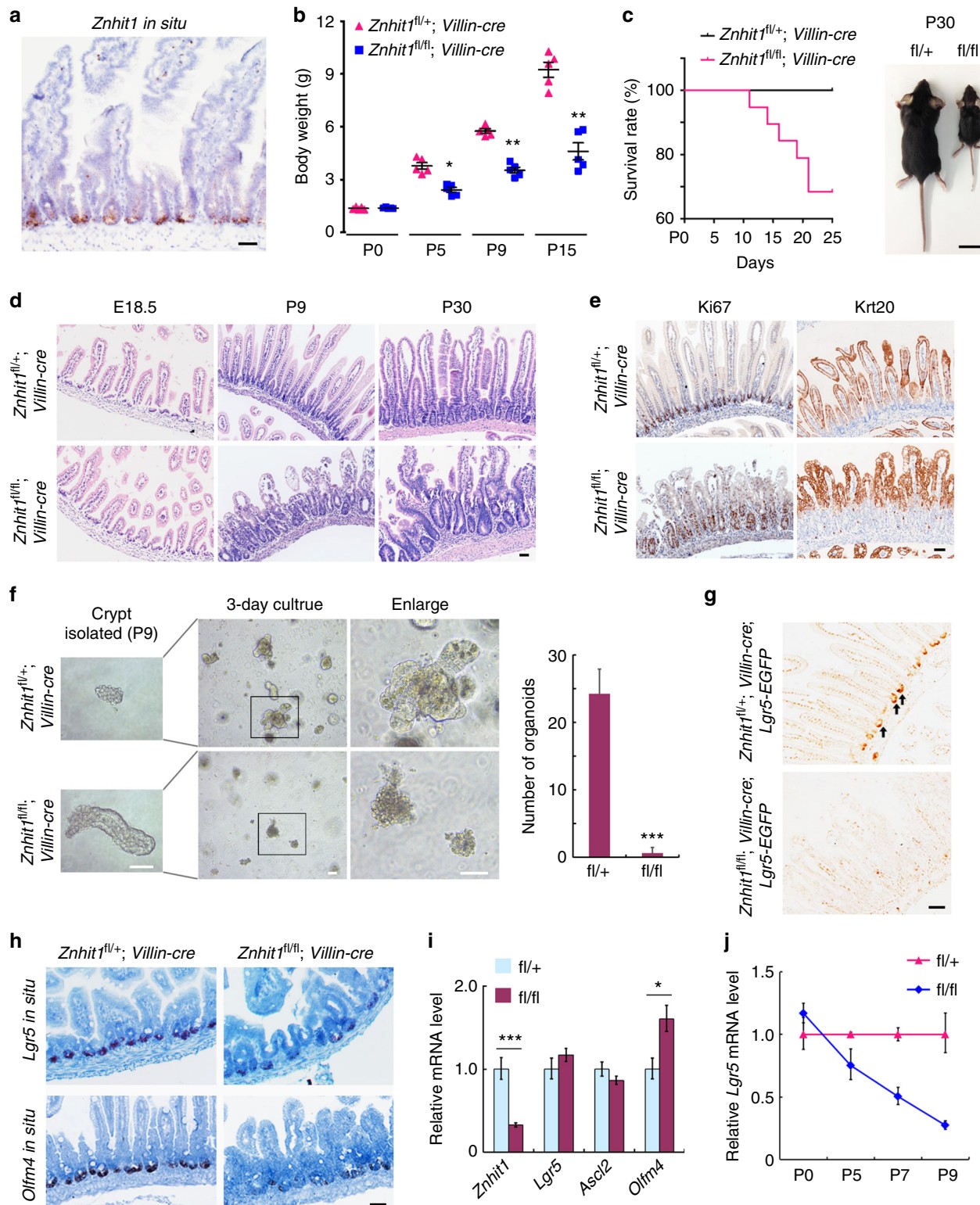

**Znhit1 is essential for Lgr5+ ISC maintenance.** Next, we examined the roles of Znhit1 in adult intestinal epithelium homeostasis maintenance by generated Znhit1 inducible knock-out mice. Four-day tamoxifen administration followed by 7-day waiting period extinguished Znhit1 in 2-month-old mice intestinal epithelium (Supplementary Fig. 4a), which led to dramatic body weight decrease and consequent individual death (Fig. 2a,

b). Histological analysis revealed the enlarged crypts and defective villi in Znhit1-deficient intestinal epithelium (Supplementary Fig. 4b, c). Furthermore, Lgr5+ ISCs were removed from adult intestinal crypts after Znhit1 deletion (Fig. 2c), which caused a loss of crypt function in generating organoids (Fig. 2d).

We then set out to examine whether Znhit1 maintains Lgr5+ ISCs in a cell-autonomous manner. Znhit1 was specifically

**Fig. 1** Znhit1 deletion disrupts postnatal generation of Lgr5+ ISC. **a** *Znhit1* in situ was performed in intestinal section of 8-week-old C57BL/6 mouse. Scale bar, 50 μm. **b** Body weight comparison between *Znhit1*[fl/+]; *Villin-cre* and *Znhit1*[fl/fl]; *Villin-cre* mice at indicated time. The data represent mean ± s.d. (*n* = 5 mice per group). Wilcoxon's rank sum test: **$P$ < 0.01. *$P$ < 0.05. **c** Kaplan–Meier survival curves of *Znhit1*[fl/+]; *Villin-cre* and *Znhit1*[fl/fl]; *Villin-cre* mice (*n* = 19 mice per genotype) and body size comparison between survived mice at P30. Scale bar, 2 cm. **d** Paraffin-embedded intestine tissues were stained with hematoxylin and eosin. **e** Ki67 and Krt20 staining of intestinal sections from *Znhit1*[fl/+]; *Villin-cre* and *Znhit1*[fl/fl]; *Villin-cre* mice at P9. **f** Intestinal crypts were isolated from *Znhit1*[fl/+]; *Villin-cre* and *Znhit1*[fl/fl]; *Villin-cre* mice at P9, embedded in Matrigel (100 crypts per well) and cultured for 3 days. The statistical analysis of organoid numbers (*n* = 5 mice per genotype) was shown as mean ± s.d. Student's *t*-test: ***$P$ < 0.001. **g** GFP staining of intestinal sections from *Znhit1*[fl/+]; *Villin-cre*; *Lgr5-EGFP-IRES-creERT2* and *Znhit1*[fl/fl]; *Villin-cre*; *Lgr5-EGFP-IRES-creERT2* mice at P9. Arrows: Lgr5+ ISCs. **h** *Lgr5* and *Olfm4* in situ were performed in intestinal sections at P9. **i** Intestine was harvested from *Znhit1*[fl/+]; *Villin-cre* (fl/+) and *Znhit1*[fl/fl]; *Villin-cre* (fl/fl) mice at P0 to examine the expression of *Znhit1*, *Lgr5*, *Ascl2*, and *Olfm4* using qRT-PCR. **j** Intestine was harvested from *Znhit1*[fl/+]; *Villin-cre* (fl/+) and *Znhit1*[fl/fl]; *Villin-cre* (fl/fl) mice at indicated time to examine *Lgr5* expression using qRT-PCR. For qRT-PCR, histone H3 was used as an internal control. The statistical data represent mean ± s.d. (*n* = 3 mice per genotype). Student's *t*-test: ***$P$ < 0.001. *$P$ < 0.05. All images are representative of *n* = 3 mice per genotype. Scale bar, 50 μm

deleted in Lgr5+ ISCs employing *Znhit1*[fl/fl]; *Olfm4-IRES-eGFPcreERT2* mice[38], in which the behavior of Lgr5+ ISCs could be easily followed by eGFP fluorescence. Indeed, 3-day tamoxifen administration followed by 4-day waiting period led to the restriction of Lgr5+ ISCs (Fig. 2e). Western blot quantification showed a significant decrease of *Olfm4*-derived eGFP protein level in intestinal crypts (Fig. 2f), indicating inhibited stemness of Lgr5+ ISCs. Consistently, Lgr5+ ISC-specific deletion of Znhit1 led to impaired organoid generating ability of intestinal crypts (Fig. 2g). Moreover, the *Znhit1*[fl/fl]; *Olfm4-IRES-eGFPcreERT2* mice subjected to long-term tamoxifen administration showed body weight decrease and intestinal epithelium transformation (Fig. 2h and Supplementary Fig. 5), which well mimicked the phenotype of *Villin-creERT*-mediated entire epithelium deletion. Taken together, these results demonstrate that Znhit1 plays critical roles in intestinal homeostasis establishment and maintenance through supporting Lgr5+ ISCs.

**Znhit1 determines the fate of Lgr5+ ISC.** To understand the underlying mechanisms of how Znhit1 supports Lgr5+ ISC and crypt function, we isolated intestinal crypts from 2-month-old *Villin-creERT* and *Znhit1*[fl/fl]; *Villin-creERT* mice at day 11 after tamoxifen administration. These intestinal crypts were subjected to examination of the gene expression profiles. RNA-sequencing revealed that Znhit1 deletion induced a significant down-regulation of 15 Lgr5+ ISC signature genes[36] (Supplementary Fig. 6), including *Lgr5*, *Olfm4*, *Clic6*, *Dach1*, *Esrrg*, and *Scn2b* (Fig. 3a, qPCR verification in Fig. 3b). While several other well-characterized Lgr5+ ISC signature genes, such as *Ascl2*, *Msi1*, and *Cdk6*, were not affected by Znhit1 deletion (Supplementary Fig. 7a). Moreover, we observed a dramatic upregulation of two critical TGF-β signaling mediators, *Tgfb1* and *Tgfbr2*[39], in Znhit1-deficient crypts (Fig. 3a, b), which was confirmed at protein level by immunostaining (Fig. 3c). Consistently, Lgr5+ ISC-specific Znhit1 deletion also resulted in the downregulation of *Lgr5*, *Olfm4*, and *Clic6* while the upregulation of *Tgfb1* and *Tgfbr2* (Fig. 3d), supporting the idea that the expression of these fate-determining genes is under control of Znhit1 in Lgr5+ ISCs.

*Lgr5* is well characterized as a Wnt target gene specifically expressed in Lgr5+ ISCs, which directly mediates their fate determination[40–44]. Therefore, specific downregulation of *Lgr5* expression might contribute to Lgr5+ ISCs depletion caused by Znhit1 deletion. Notably, there was no change of either *Ascl2* (Supplementary Fig. 7a), which is a Wnt-targeted master transcription factor activating *Lgr5* transcription[45], or *Axin2* (Supplementary Fig. 7a, b), which is a classic Wnt signaling activity indicator[46], suggesting that Znhit1 regulates *Lgr5* gene expression without affecting Wnt signaling. In addition, increased expression of Tgfb1 and Tgfbr2 resulted in enhanced phosphorylation of Smad2 (Fig. 3e and Supplementary Fig. 8), which is an

indicator of TGF-β signaling activity[39,47], suggesting that TGF-β activation might also participate in the negative effect of Znhit1 deficiency on Lgr5+ ISCs. Remarkably, reconstitution of Lgr5 expression through hyperactivating Wnt signaling with CHIR99021[48,49] and inhibition of TGF-β signaling activity with SB431542[50] could cooperate to rescue the crypt dysfunction caused by ISC-specific Znhit1 deletion (Fig. 3f). The ablated robust expression of *Lgr5* and *Olfm4* was reestablished as well (Fig. 3g), supporting the idea that both Lgr5 suppression and TGF-β activation are involved in Lgr5+ ISCs depletion after Znhit1 deletion.

As previous studies revealed a critical role of TGF-β signaling in Paneth cells differentiation[13], we also examined the presence of Paneth cells in Znhit1-deficient crypts. Indeed, Znhit1 deficiency led to the expansion of Paneth population, illustrated by the upregulation of Paneth cell markers *Pla2g2e* and *Lyz2* (Fig. 3a, b, d) and increased lysozyme staining (Supplementary Fig. 9). These data are consistent with our observation of upregulated TGF-β signaling activity in the Znhit1 mutant mice.

**Znhit1 incorporates H2A.Z for transcriptional regulation.** As a component of SRCAP complex, Znhit1 has been shown to regulate histone variant H2A.Z deposition, thereby controlling gene transcription[26,31,35]. To determine whether Znhit1 regulates the transcription of fate-determining genes in intestinal epithelium through altering the H2A.Z deposition at the gene loci, we performed H2A.Z genome-wide chromatin immunoprecipitation and sequencing (ChIP-seq) analysis on wild-type and Znhit1-deficient crypts. ChIP-seq data revealed 6506 H2A.Z-binding sites in wild-type crypts, of which, 88.55% were within the annotated gene regions, including promoter-proximal (−5 to −0.5 kb from TSS), TSS region (−0.5 to +0.5 kb from TSS), and non-TSS exon and intron (Fig. 4a). Of note, more than half of the peaks (3859 peaks, 59.3%) were restricted in TSS regions, which is in consistent with previous reports showing the association of H2A.Z with functional regulatory elements close to TSS in mammalian genome[29,30,51]. Furthermore, most of the H2A.Z-binding sites, especially the TSS-located ones, were removed from genome by Znhit1 deletion (*Ereg* and *Fbp1* loci were shown as examples) (Fig. 4a), demonstrating that Znhit1 is essential for global H2A.Z deposition.

The TSS-located H2A.Z-binding sites, which have the largest potential in regulating gene transcription, were annotated to a total of 3698 genes. Comparing them with 949 Znhit1-regulated genes identified in RNA-seq, we found that 107 Znhit1-regulated genes had H2A.Z occupation in TSS regions (Fig. 4b and gene list in Supplementary Table 1). GO analysis indicated the enrichment of regulation of cell proliferation, pathway-restricted SMAD protein phosphorylation, and organ regeneration (Supplementary Fig. 10). Remarkably, the four Znhit1-regulated fate-determining

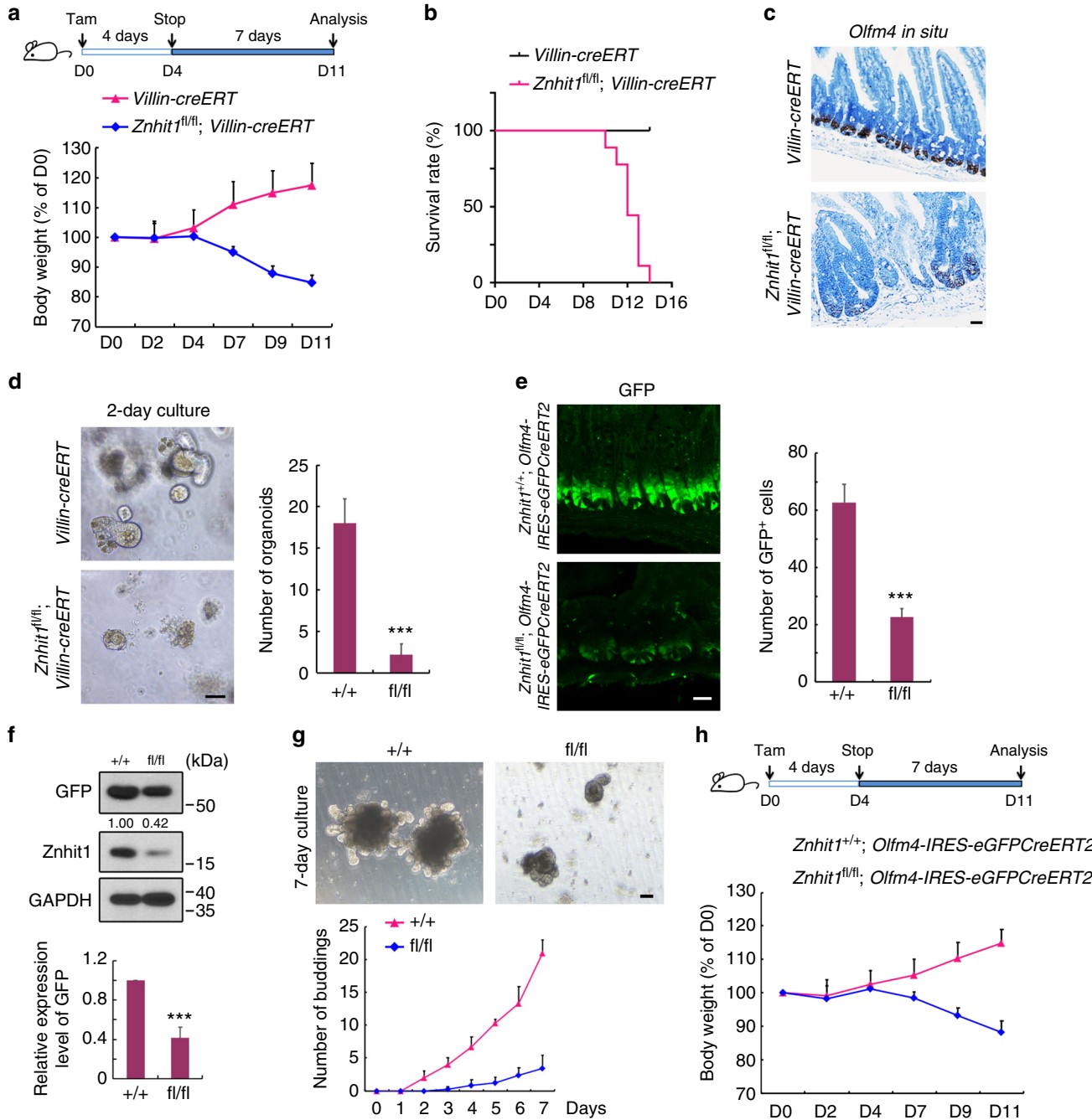

**Fig. 2** Znhit1 is essential for Lgr5+ ISC maintenance thus intestinal homeostasis. **a** Eight-week-old *Villin-creERT* and *Znhit1*fl/fl; *Villin-creERT* mice were daily injected with tamoxifen for 4 days followed by 7-day waiting period. Top: Scheme of Cre induction strategy. Bottom: Body weight comparison between *Villin-creERT* and *Znhit1*fl/fl; *Villin-creERT* mice at indicated time following tamoxifen treatment. (*n* = 5 mice per genotype). **b** Kaplan–Meier survival curves of *Villin-creERT* and *Znhit1*fl/fl; *Villin-creERT* mice post tamoxifen administration (*n* = 9 mice per genotype). **c** *Olfm4* in situ was performed in intestinal sections from *Villin-creERT* and *Znhit1*fl/fl; *Villin-creERT* mice following tamoxifen treatment. **d** Intestinal crypts were isolated from *Villin-creERT* and *Znhit1*fl/fl; *Villin-creERT* mice following tamoxifen treatment, embedded in Matrigel (100 crypts per well) and cultured for 2 days. The statistical analysis of organoid numbers (*n* = 3 mice per genotype) was shown. **e** Eight-week-old *Znhit1*+/+; *Olfm4-IRES-eGFPCreERT2* and *Znhit1*fl/fl; *Olfm4-IRES-eGFPCreERT2* mice were daily injected with tamoxifen for 3 days followed by 4-day waiting period, then Lgr5+ ISCs were examined by confocal cross-sectioning. The GFP+ cells were quantified for statistical analysis (*n* = 3 mice per genotype). **f** Intestinal crypts were isolated from *Znhit1*+/+; *Olfm4-IRES-eGFPCreERT2* (+/+) and *Znhit1*fl/fl; *Olfm4-IRES-eGFPCreERT2* (fl/fl) mice following tamoxifen treatment for immunoblotting with the indicated antibodies. GAPDH served as a loading control. The statistical data represent mean+ s.d. (*n* = 3 mice per genotype). **g** Intestinal crypts were isolated from *Znhit1*+/+; *Olfm4-IRES-eGFPCreERT2* (+/+) and *Znhit1*fl/fl; *Olfm4-IRES-eGFPCreERT2* (fl/fl) mice following tamoxifen treatment, embedded in Matrigel (100 crypts per well) and cultured for 7 days. The organoid buddings were quantified for statistical analysis (*n* = 5 mice per genotype). **h** *Znhit1*+/+; *Olfm4-IRES-eGFPCreERT2* and *Znhit1*fl/fl; *Olfm4-IRES-eGFPCreERT2* mice were daily injected with tamoxifen for 4 days followed by 7-day waiting period. Body weight comparison was shown (*n* = 3 mice per genotype). The statistical data represent mean ± s.d. Student's *t*-test: ***$P$ < 0.001. All images are representative of *n* = 3 mice per genotype. Scale bar, 50 μm

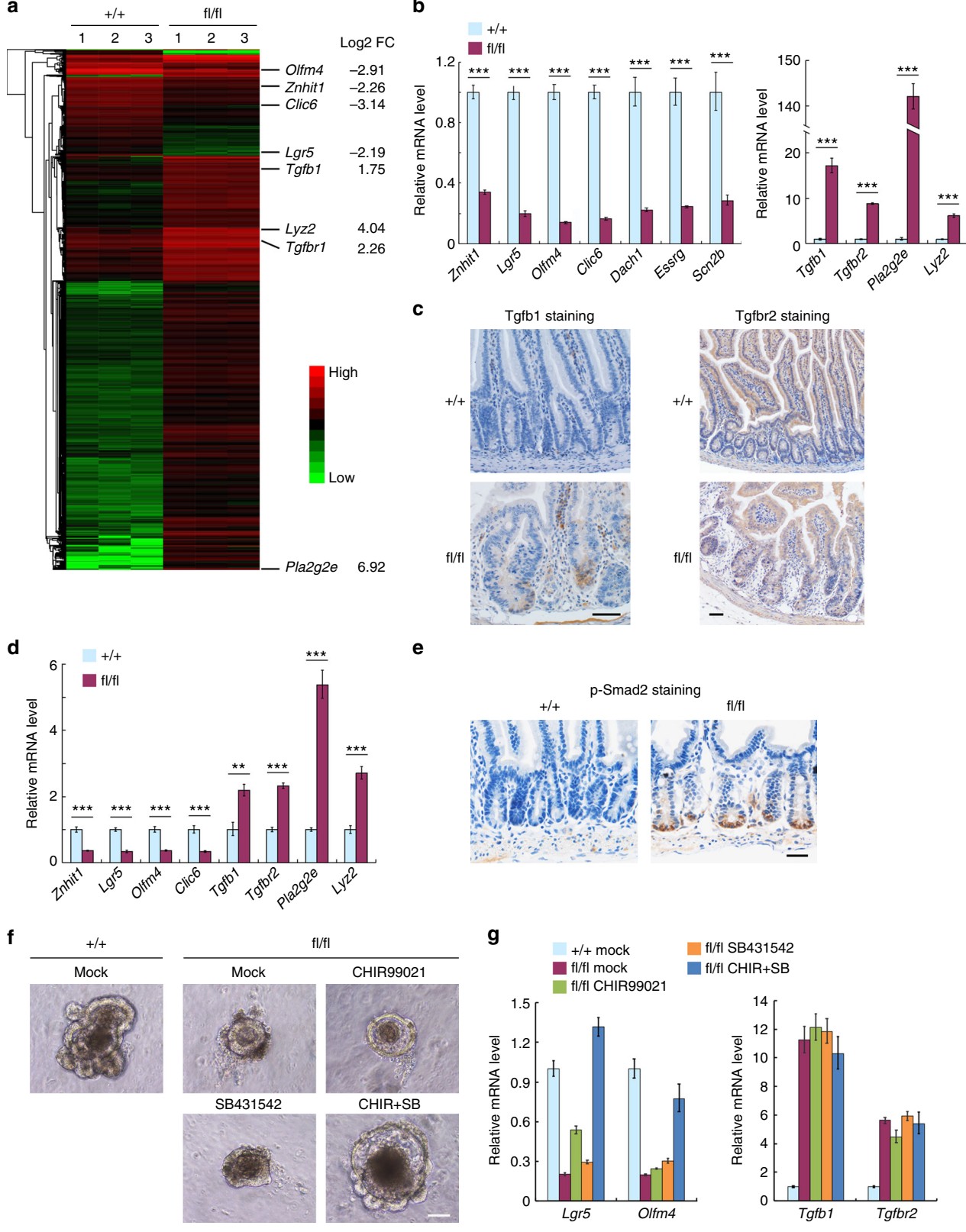

genes, *Lgr5*, *Clic6*, *Tgfb1*, and *Tgfbr2*, all had prominent H2A.Z ChIP-seq signals in their TSS regions, which could not be detected after Znhit1-deletion (Fig. 4c). ChIP-qPCR further confirmed the removal of H2A.Z incorporation into TSS regions of *Lgr5*, *Clic6*, *Tgfb1*, and *Tgfbr2* caused by Znhit1 deficiency

(Fig. 4d). These data suggest that Znhit1 might directly regulate the transcription of these genes through mediating H2A.Z TSS incorporation.

We employed H2A.Z conditional knockout mice to assess further the significance of H2A.Z in transcription regulation of

**Fig. 3** Znhit1 controls the transcription of Lgr5+ ISC fate-determining genes. **a**, **b** Eight-week-old *Villin-creERT* (+/+) and *Znhit1*^fl/fl^; *Villin-creERT* (fl/fl) mice were daily injected with tamoxifen for 4 days followed by 7-day waiting period. Intestinal crypts were harvested for RNA-seq (**a**) and qRT-PCR (**b**) to analyze the gene expression changes. Clustered heatmap of log2-transformed RPKMs shows the differentially expressed genes after Znhit1 deletion. Log2-transformed fold changes of indicated genes were marked in right. **c** Tgfb1 and Tgfbr2 staining of intestinal sections from *Villin-creERT* (+/+) and *Znhit1*^fl/fl^; *Villin-creERT* (fl/fl) mice following tamoxifen treatment. **d** Eight-week-old *Znhit1*^+/+^; *Olfm4-IRES-eGFPCreERT2* (+/+) and *Znhit1*^fl/fl^; *Olfm4-IRES-eGFPCreERT2* (fl/fl) mice were daily injected with tamoxifen for 3 days followed by 4-day waiting period. Intestinal crypts were harvested to examine the expression of indicated genes using qRT-PCR. **e** Phospho-Smad2 staining of intestinal sections from *Villin-creERT* (+/+) and *Znhit1*^fl/fl^; *Villin-creERT* (fl/fl) mice following tamoxifen treatment. **f** Intestinal crypts were isolated from *Znhit1*^+/+^; *Olfm4-IRES-eGFPCreERT2* (+/+) and *Znhit1*^fl/fl^; *Olfm4-IRES-eGFPCreERT2* (fl/fl) mice following tamoxifen treatment and subjected to in vitro culture in the presence of 3 μM CHIR99021 and/or 10 μM SB431542 for 4 days. Mock: DMSO. **g** The cultured organoids were harvested to examine the expression of indicated genes using qRT-PCR. For qRT-PCR, histone H3 was used as an internal control. The statistical data represent mean ± s.d. (n = 3 mice per genotype or treatment). Student's *t*-test: \*\*\*P < 0.001. \*\*P < 0.01. All images are representative of n = 3 mice per genotype. Scale bar, 50 μm

these fate-determining genes. H2A.Z has two isoforms in mouse, *H2afv* and *H2afz*, which are encoded by separated loci[52]. Consistently, knockout of both of the two isoforms, efficiently suppressed the expression of *Lgr5* and *Clic6*, while enhanced the expression of *Tgfb1* and *Tgfbr2* (Fig. 4e). Moreover, *H2afv* and *H2afz* double knockout, but not single isoform deletion, led to dramatic body weight decrease, intestinal epithelium transformation and crypt dysfunction (Supplementary Fig. 11a–c), which well mimicked the Znhit1-deficient defects. Taken together, we demonstrate that Znhit1 regulates the expression of fate-determining genes through mediating H2A.Z incorporation into their TSS regions.

To investigate how TSS H2A.Z incorporation regulates gene transcription, we performed ChIP-qPCR to examine the histone H3 epigenetic modification landmarks at different loci in wild-type and H2A.Z-deficient crypts. We found that *Lgr5* TSS region had an original enrichment of H3K4me3 (transcription activation landmark), while *Tgfb1* TSS region had an original enrichment of H3K27me3 (transcription suppression landmark) (Fig. 4f). Both H3K4me3 and H3K27me3 landmarks were efficiently ablated after H2A.Z deletion (Fig. 4f), suggesting that the histone H3 methylation status might determine the opposite regulatory effects of H2A.Z on transcription of different genes (upregulation of *Lgr5* and *Clic6*, while downregulation of *Tgfb1* and *Tgfbr2*). Interestingly, as H2A.Z deficiency did not disrupt the balance between H3K4me3 and H3K27me3 on TSS region of *Mettl3* or *Prmt1* (Fig. 4f), the transcription was not affected (Fig. 4e). These data together suggest that H2A.Z specifically controls gene transcription through permitting regulatory histone H3 methylations.

**Znhit1 enhances the interaction between H2A.Z and YL1.** We further explored the molecular mechanism of how Znhit1 mediates H2A.Z incorporation. As it is unknown how Znhit1 functions in SRCAP complex, we first examine whether Znhit1 deficiency can affect the stability or modification of other key components. Interestingly, we observed a novel form of YL1 above the main band in wild-type but not Znhit1-deficient intestinal crypts (Fig. 5a). This form of YL1 could be detected by anti-phospho-serine/threonine (p-Akt substrate) antibody (Fig. 5b) and was sensitive to shrimp alkaline phosphatase (rSAP) treatment (Fig. 5c), indicating that it is a phosphorylated form of YL1. Importantly, immunoprecipitation revealed that Znhit1 deletion efficiently abolished the interaction between p-Akt and YL1 without affecting Akt activity (Fig. 5d), indicating that Znhit1 is essential for the binding of p-Akt to YL1 and consequent YL1 phosphorylation.

Recent studies show that YL1 directly interacts with H2A.Z and mediates its deposition[33,34]. Of note, the phosphorylated YL1 showed stronger affinity with H2A.Z than non-

phosphorylated YL1 (Fig. 5e). Ablating YL phosphorylation in crypt lysis by rSAP or in cultured organoids by PI3K-Akt inhibitor LY294002 eliminated the binding of YL1 to H2A.Z (Fig. 5f, g). These data suggest that Znhit1 might promote the binding of YL1 to H2A.Z through supporting YL1 phosphorylation. Indeed, Znhit1 interacted with YL1 (Fig. 5f) and its deletion efficiently abolished the interaction between H2A.Z and YL1 (Fig. 5h). Taken together, we demonstrate that Znhit1 maintains the interaction between H2A.Z and YL1 through enhancing YL1 phosphorylation.

## Discussion

Lgr5+ ISCs play a dominant role in maintaining gastrointestinal epithelium homeostasis. As the population abnormality has been directly linked to intestinal epithelium degeneration and colonic tumorogenesis[12,53], understanding the mechanism of how Lgr5+ ISCs are generated and maintained would provide potential therapeutic intervention of gastrointestinal diseases[4,54]. Here, we examine the critical roles of Znhit1 in intestinal epithelium homeostasis establishment and maintenance and demonstrate that Znhit1 is essential for Lgr5+ ISC self-renewal. Our experimental data favor a model for Znhit1 in determining the fate of Lgr5+ ISCs: Znhit1 promotes the interaction between H2A.Z and YL1 thus mediates H2A.Z incorporation into TSS region of *Lgr5*, *Tgfb1*, and *Tgfbr2* for transcriptional regulation; Znhit1 deficiency leads to Lgr5 suppression and TGF-β signaling activation, which drive the self-renewal to differentiation transition of Lgr5+ ISCs (Fig. 5h). Previous studies of Lgr5+ ISC fate determination mainly focused on functions of signaling pathways and their downstream effectors. Thus, to our knowledge, our study at the first elucidate how chromosome remodeling factors control gene expression pattern and thus Lgr5+ ISC fate determination.

Znhit1 deletion depletes Lgr5+ ISC from P9 intestinal epithelium, demonstrating Znhit1 is essential for Lgr5+ ISC postnatal generation. Notably, we observed that normal intestinal epithelium underwent rapid crypt fission to increase the number of crypts at P9, which was remarkably blocked by Znhit1 deficiency. Previous studies recognized the expansion of ISCs or progenitors as the major cause of crypt fission[3], supporting the idea that Znhit1 deficiency-caused Lgr5+ ISC depletion leads to crypt fission failure thus morphological enlarged crypts. Moreover, in consistent with our observation of Paneth population expansion after Znhit1 deletion, Garcia et al. reported that Lgr5 deficiency could lead to premature Paneth cell differentiation in the small intestine[22], suggesting Znhit1 deletion might also contribute to secretory lineage differentiation through restricting Lgr5+ ISCs. These results agree with our conclusion that Lgr5+ ISC depletion is the primary defect of Znhit1 deletion in intestinal epithelium.

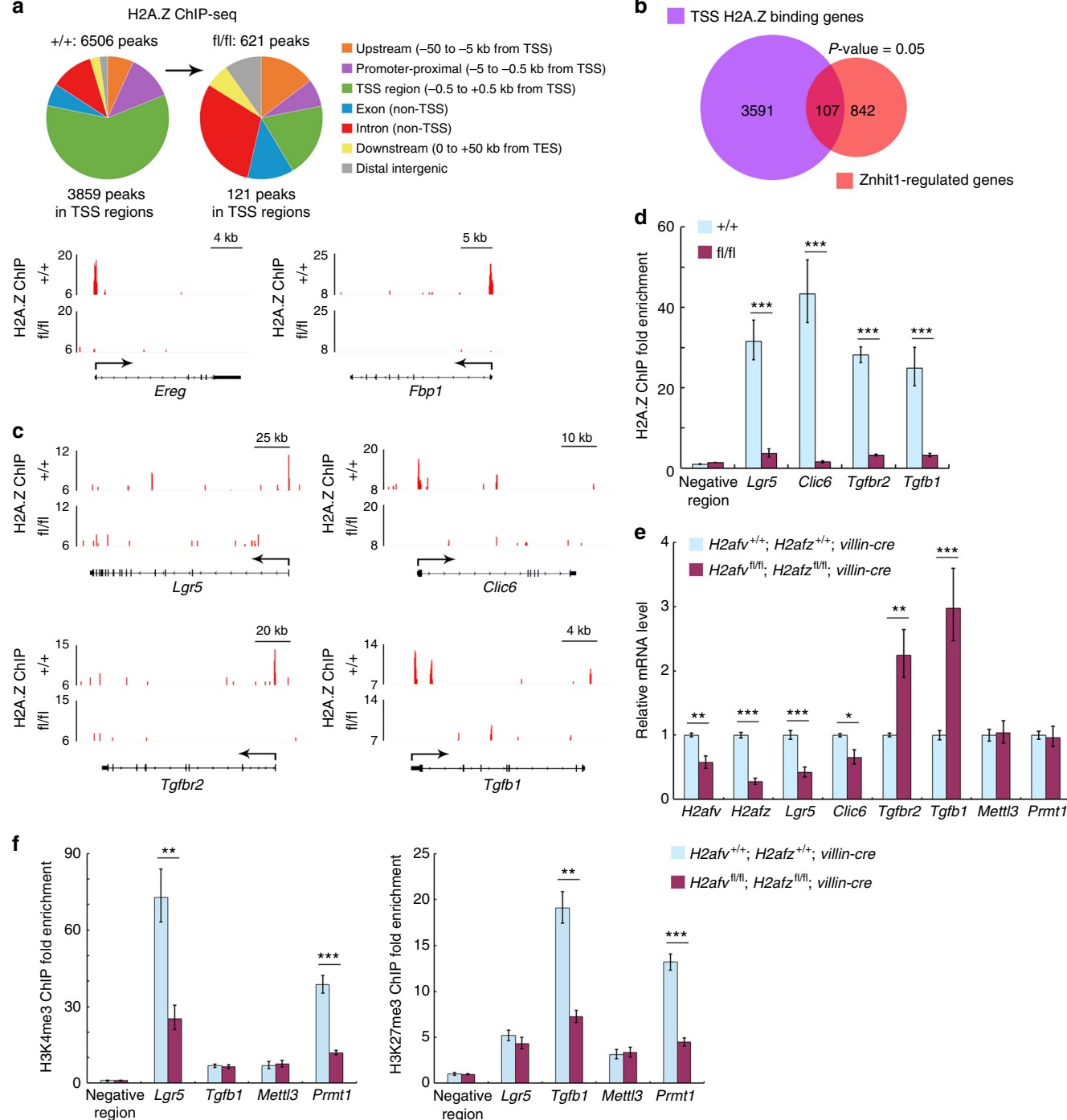

**Fig. 4** Znhit1 incorporates H2A.Z for transcriptional regulation. **a** Distribution of H2A.Z on genome of intestinal crypts. Eight-week-old *Villin-creERT* (+/+) and *Znhit1*fl/fl; *Villin-creERT* (fl/fl) mice were daily injected with tamoxifen for 3 days followed by 5-day waiting period. Intestinal crypts were harvested for ChIP-seq. ChIP-seq signals for H2A.Z binding at *Ereg* and *Fbp1* loci were shown as examples. **b** Venn diagram showing the overlap between TSS H2A.Z binding genes and Znhit1-regulated genes. The significance was evaluated by Fisher's exact test. **c** ChIP-seq signals for H2A.Z binding at *Lgr5*, *Clic6*, *Tgfbr2*, and *Tgfb1* loci. **d** Eight-week-old *Villin-creERT* (+/+) and *Znhit1*fl/fl; *Villin-creERT* (fl/fl) mice were daily injected with tamoxifen for 3 days followed by 5-day waiting period. Intestinal crypts were harvested and ChIP-qPCR was performed to examine the fold enrichment of H2A.Z in TSS region of indicated genes. **e** Intestinal crypts were harvested from *H2afv*+/+; *H2afz*+/+; *Villin-cre* and *H2afv*fl/fl; *H2afz*fl/fl; *Villin-cre* mice at P9 to examine the expression of indicated genes using qRT-PCR. Histone H3 was used as an internal control. **f** Intestinal crypts were harvested from *H2afv*+/+; *H2afz*+/+; *Villin-cre* and *H2afv*fl/fl; *H2afz*fl/fl; *Villin-cre* mice at P9 and ChIP-qPCR was performed to examine the fold enrichment of H3K4me3 and H3K27me3 in TSS region of indicated genes. The statistical data represent mean ± s.d. (n = 3 mice per genotype). Student's t-test: ***P < 0.001. **P < 0.01. *P < 0.05

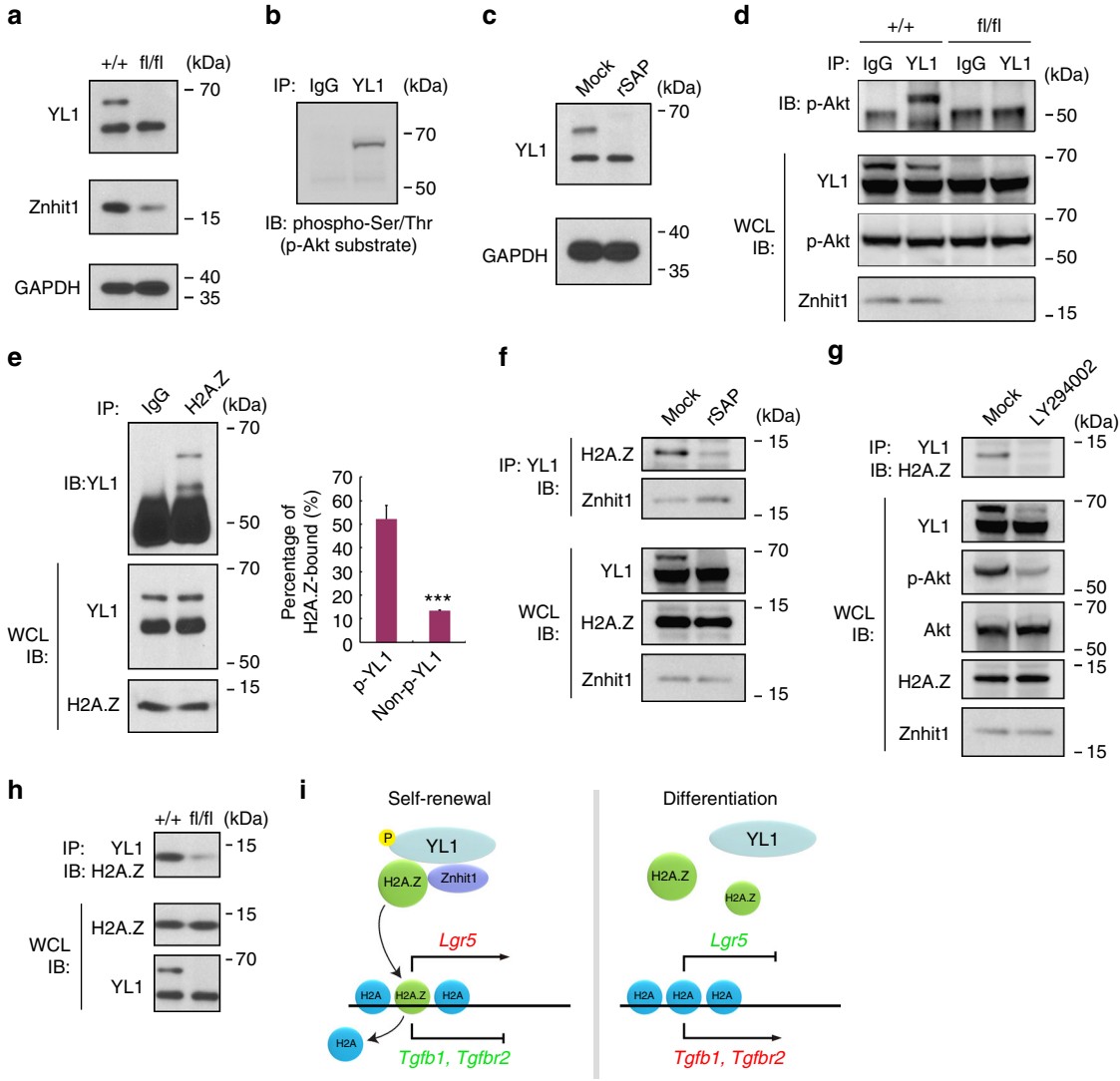

**Fig. 5** Znhit1 mediates H2A.Z incorporation by enhancing the interaction between H2A.Z and YL1. **a** Eight-week-old *Znhit1*[+/+]; *Olfm4-IRES-eGFPCreERT2* (+/+) and *Znhit1*[fl/fl]; *Olfm4-IRES-eGFPCreERT2* (fl/fl) mice were daily injected with tamoxifen for 3 days followed by 4-day waiting period, then intestinal crypts were isolated for immunoblotting with the indicated antibodies. GAPDH served as a loading control. **b** Intestinal crypt (wild-type) were harvested for anti-YL1 immunoprecipitation then anti-phospho-Ser/Thr (p-Akt substrate) immunoblotting. **c** Intestinal crypt (wild-type) lysis was treated with rSAP for dephosphorylation then subjected to immunoblotting with the indicated antibodies. **d** Intestinal crypts were harvested from *Znhit1*[+/+]; *Olfm4-IRES-eGFPCreERT2* (+/+) and *Znhit1*[fl/fl]; *Olfm4-IRES-eGFPCreERT2* (fl/fl) mice following tamoxifen treatment for anti-YL1 immunoprecipitation then anti-p-Akt immunoblotting. WCL IB: whole cell lysis immunoblotting. **e** Intestinal crypts (wild-type) were harvested for anti-H2A.Z immunoprecipitation then anti-YL1 immunoblotting. Percentage of H2A.Z-bound phospho-YL1 (p-YL1) and non-phospho-YL1 (non-p-YL1) was quantitated. The statistical data represent mean ± s.d. ($n = 3$ mice). Student's $t$-test: ***$P < 0.001$. **f** Intestinal crypt (wild-type) lysis treated with rSAP were subjected to anti-YL1 immunoprecipitation then immunoblotting with the indicated antibodies. **g** Cultured organoids (wild-type) were treated with 25 μM LY294002 for 12 h then harvested for anti-YL1 immunoprecipitation followed by immunoblotting. **h** Intestinal crypts were harvested from *Znhit1*[+/+]; *Olfm4-IRES-eGFPCreERT2* (+/+) and *Znhit1*[fl/fl]; *Olfm4-IRES-eGFPCreERT2* (fl/fl) mice following tamoxifen treatment for anti-YL1 immunoprecipitation then anti-H2A.Z immunoblotting. **i** Working model of how Znhit1-mediated H2A.Z incorporation regulates the transcription of Lgr5+ ISC fate determiners

To date, due to the lack of genetic mice tool, the in vivo function of Znhit1 is unknown. Our studies pinpoint the critical roles of Znhit1 in regulating Lgr5+ ISC activities and intestinal homeostasis, providing the first description of Znhit1 in vivo function. We further define Znhit1 as a H2A.Z global incorporation mediator in intestinal crypts and demonstrate Znhit1 exerts its function through H2A.Z, which uncover the transcription-regulating effect of H2A.Z on fate-determining genes in Lgr5+ ISCs. Worth to mention, although the functions of H2A.Z in regulating chromosome structure and gene expressions have been documented from yeast to human[26], most

mammalian studies were performed in cultured cell lines[28–30,35,55–57]. Our studies determine a dominant role of H2A.Z in mammalian organ development and tissue homeostasis.

To reveal a global map of H2A.Z distribution on the genome of mouse intestinal crypts, we performed anti-H2A.Z ChIP-seq experiment. In consistent with previous anti-H2A.Z ChIP-seq performed in ESC[29,30], H2A.Z is highly accumulated at TSS region of a large group of genes. Interestingly, although Znhit1 deletion removes most of the H2A.Z peaks, only 107 genes have dramatic transcriptional change, indicating H2A.Z selectively regulates a small part of the bound genes. Further experiments are

needed to address the mechanism of how this selectivity is built up timely and spatially, and how the epigenetic modifications of DNA and histone work together on H2A.Z-bound gene promoters to precisely control the expression of essential developmentally regulated genes.

Besides investigating the function of H2A.Z in gene expression regulation, it is important to elucidate the molecular mechanism of how H2A.Z is incorporated into chromatin. Recent studies identified YL1 as the first specific H2A.Z-deposition chaperone. However, it is unknown whether the interaction between H2A.Z and YL1 can be regulated. Here, we demonstrate that Znhit1 mediates the global incorporation of H2A.Z through promoting the interaction between H2A.Z and YL1 in intestinal crypts. At the same time, we discover a Znhit1-dependent phosphorylation of YL1, which directly controls its binding ability to H2A.Z. These results greatly support the idea that H2A.Z incorporation is dynamically regulated by Znhit1 and YL1. To further evaluate the significance of YL1 phosphorylation in H2A.Z incorporation, the phosphorylation site and involved kinase should be determined.

In summary, by employing conditional knockout mice and taking intestinal epithelium as a research model, we address a dominant role of Znhit1/H2A.Z in Lgr5+ ISC maintenance and intestinal homeostasis, which leads the investigation of how Znhit1/H2A.Z controls mammalian organ development and tissue homeostasis in vivo.

## Methods

**Mice**. Znhit1<sup>fl/fl</sup> mice were generated by Model Animal Research Center of Nanjing University (MARC, Nanjing, China). The targeting strategy is shown in Supplementary Fig. 2a. Genotyping primer pair: sense 5′-GTTGGGAGCATCTGCCT TTC-3′, anti-sense 5′-CCCTGCCTACATCTGCACTAA-3′.

Villin-cre, Villin-creERT, and Lgr5-EGFP-IRES-creERT2 mice were obtained from the Jackson Laboratory. Olfm4-IRES-eGFPcreERT2 mice were provided by Hans Clevers. H2afv<sup>fl/fl</sup>/H2afz<sup>fl/fl</sup> mice were obtained from RIKEN BioResource Center. All strains were maintained in C57BL/6 background. For Cre induction, mice were intraperitoneally injected with 100 μl tamoxifen in sunflower oil at 20 mg/ml for 3–4 consecutive days.

All breeding and experimental procedures were performed in accordance with the relevant guidelines and regulations and with the approval of the Institutional Animal Care and Use Committee at Fudan University or Cincinnati Children's Hospital.

**Reagents**. Matrigel was purchased from BD Biosciences, recombinant mouse Noggin, recombinant human R-Spondin1 from R&D Systems, recombinant mouse EGF, advanced DMEM/F12, penicillin/streptomycin, GlutaMAX-I, N2, and B27 from Invitrogen, N-Acetylcysteine, BSA, EDTA, CHIR99021, SB431542, and LY294002 were from Sigma, and rSAP from New England Biolabs.

**Immunohistochemistry**. Tissues were fixed with 4% paraformaldehyde and embedded in paraffin. Sections were deparaffinized in xylene and graded alcohols, followed by antigen retrieval, and endogenous peroxidase quenched by $H_2O_2$. Sections were then blocked with 1% BSA in PBS for 30 min, and incubated overnight at 4 °C with α-Ki67 (BD-550609, 1:200), α-Krt20 (CST-13063, 1:300), α-Tgfb1 (Santa Cruz-sc130348, 1:300), α-Tgfbr2 (Santa Cruz-sc17792, 1:50), α-phospho-Smad2 (CST-18338, 1:100), or α-Lysozyme (Santa Cruz-sc27958, 1:200). Secondary biotinylated α-mouse IgG or α-rabbit IgG (Invitrogen, 1:5000) was added for 30 min, followed by detection with streptavidin-HRP and DAB+ chromogen (Invitrogen) according to the manufacturer's recommendations. Slides were counterstained with Mayer's hematoxylin, dehydrated, and mounted with Eukitt (Sigma). Images were taken by Vectra Automated Quantitative Pathology Imaging System (Perkin Elmer).

**Isolation of intestinal crypts and organoid culture**. Mouse intestine was isolated, cut longitudinally, and washed twice with cold PBS. Villi were carefully scraped off with operating scalpel. The remaining part was cut into small pieces (5 mm) and incubated in 10 mM EDTA in PBS for 40 min on ice. After removal of EDTA, the small pieces were vigorously suspended using a 10-ml pipette with cold PBS. The supernatant, which enriched in crypts, was passed through 70 μm cell strainer (BD) and centrifuged at 600 rpm for 3 min. The crypts obtained were embedded in Matrigel, followed by seeding on a 48-well plate. After polymerization of Matrigel, ENR crypt culture medium (advanced DMEM/F12 supplemented with penicillin/streptomycin, GlutaMAX-I, N2, B27, and N-acetylcysteine containing 50 ng/ml

EGF, 100 ng/ml Noggin, and 500 ng/ml R-spondin1)[12] was overlaid. For chemical treatment, 3 μM CHIR99021 and 10 μM SB431542 were added in culture medium.

**In situ hybridization**. Intestines from mice were flushed with cold PBS and fixed overnight in 4% paraformaldehyde. Samples were then dehydrated and embedded in paraffin, sectioned at 5 μm, and processed to in situ hybridization with the RNA scope 2.0 kit (Advanced Cell Diagnostics).

**Quantitative RT-PCR (qRT-PCR)**. Total RNA was extracted with RNeasy Mini Kit (QIAGEN) and cDNA was prepared using GoScript Reverse Transcription System (Promega). Real-time PCR reactions were performed in triplicates on CFX96 Touch System (BioRad). Primers used are listed in Supplementary Table 2.

**RNA-seq**. RNA from freshly isolated intestinal crypts was converted into cDNA libraries using the Ovation® RNA-Seq System V2 kit (NuGEN). High-throughput sequencing was performed using Illumina HiSeq X10 for 3 biological replicates, respectively. For each sample, the RNA-seq data was mapped to mm10 genome by TopHat v1.4.1[58] with no more than 2 mismatches, and then only the uniquely mapped reads were used to estimate the expression values in gene level by RPKM[59]. Statistical significant test of differentially expressed genes was performed by DEseq with R. Genes with absolute log2-transformed fold changes greater than 1.7 were regarded as differentially expressed genes and a threshold of $p$ value < 0.01 was used. Hierarchical clustering of log2-transformed RPKMs was generated by Cluster 3.0 and visualized by Java TreeView. The raw NGS data were deposited to the NCBI SRA database under accession number (SRP148616).

**ChIP-qPCR and ChIP-seq**. Freshly isolated intestinal crypts were cross-linked with 1% formaldehyde for 10 min at room temperature, quenched with glycine, and successively washed with phosphate-buffered saline. The cells were then homogenized and resuspended in shearing buffer (1% SDS, 50 mM Tris–HCl pH 8.0, 10 mM EDTA pH 8.0) and sheared using Bioruptor Plus (Bioruptor) for 20 min with the following settings: high power, 30 s on, 30 s off, 20 cycles. For each ChIP, 100 μl of the sonicated chromatin was diluted to 0.06% SDS, incubated for 12 h at 4 °C with 2 μl α-H2A.Z (Abcam-ab4174), α-H3K4me3 (CST-9751), or α-H3K27me3 (CST-9733) antibody and 20 μl of protein A/G magnetic beads (Millipore). The beads were successively washed once with buffer 1 (50 mM Tris–HCl pH 8.0, 0.15 M NaCl, 1 mM EDTA pH 8.0, 0.1% SDS, 0.1% deoxycholate, 1% Triton X-100), two times with buffer 2 (50 mM Tris–HCl pH 8.0, 0.5 M NaCl, 1 mM EDTA pH 8.0, 0.1% SDS, 0.1% deoxycholate, 1% Triton X-100), two times with buffer 3 (50 mM Tris–HCl pH 8.0, 0.5 M LiCl, 1 mM EDTA pH 8.0, 1% Nonidet P-40, 0.7% deoxycholate), and two times with buffer 4 (10 mM Tris–HCl pH 8.0, 1 mM EDTA, pH 8.0) for 10 min at 4 °C. Chromatin was eluted by incubation of the beads with elution buffer (10 mM Tris–HCl pH 8.0, 0.3 M NaCl, 5 mM EDTA pH 8.0, 0.5% SDS, 1 μl RNaseA) for 3 h at 65 °C. After 2-h incubation with proteinase K at 55 °C, DNA was extracted with phenol–chloroform and precipitated with ethanol. ChIP-qPCR was performed with the primers listed in Supplementary Table 3. Or the immunoprecipitated chromatin was subjected to library construction and sequencing on Illumina HiSeq 2500 by BerryGenomics. Sequencing reads were aligned to the reference genome (mm10) using Bowtie v1.1.1 with no more than 2 mismatches, and then only the uniquely mapped reads were used for peak calling analysis. The peaks detection was performed by MACS with default cutoff. Peaks were assigned to the nearest genes using Homer. The raw NGS data were deposited to the NCBI SRA database under accession number (SRP148519).

**Immunoblotting and immunoprecipitation**. These assays were performed using the following antibodies: α-GFP (Clontech-8372-2, 1:2000), α-Znhit1 (Sigma-HPA019043, 1:100), α-GAPDH (Origene-TA802519, 1:2000), α-YL1 (Abcam-ab112055, 1:2000 for IB, 1:100 for IP), α-phospho-Ser/Thr (p-Akt substrate) (CST-9614, 1:1000), α-p-Akt (CST-4060, 1:2000), α-H2A.Z (Abcam-ab4174, 1:2000 for IB, 1:100 for IP), α-Akt (CST-4685, 1:2000), and ECL HRP-conjugated α-mouse IgG and α-rabbit IgG (GE Healthcare, 1:10,000). The experiments were repeated for at least three times, and representative data were shown. The band intensity was quantitated with BandScan 5.0. The full immunoblots are provided in Supplementary Fig. 12.

**Statistical analysis**. We employed Student's t-test or ANOVA test to analyze the parametric experimental results. In nonparametric data analysis, we employed Wilcoxon's rank sum test for two-group and Kruskal–Wallis' $H$ test for multi-group. Significant differences were noted with asterisks. We performed Fisher's exact test to evaluate the significant enrichment of the overlap between TSS H2A.Z binding genes and Znhit1-regulated genes.

**Reporting summary**. Further information on experimental design is available in the Nature Research Reporting Summary linked to this article.

## Data availability

The raw NGS data were deposited to the NCBI SRA database under accession number SRP148616 (RNA-seq data) and SRP148519 (ChIP-seq data). The data will be released

upon publication. All other data of this study are available from the corresponding authors upon reasonable request.

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

## Acknowledgements

The authors wish to thank Dr. Guohong Li for technical support and discussions, Dr. Hans Clevers for *Olfm4-IRES-eGFPcreERT2* mouse strain, and Dr. Wanzhu Jin for support on the *H2afv*<sup>fl/fl</sup>/*H2afz*<sup>fl/fl</sup> mouse work. This work was supported by grants from the National Natural Science Foundation of China (31730044 and 31771614) and NIH grants (R01GM115995 and R01HL136722). B.Z. was sponsored by Shanghai Rising-Star Program.

## Author contributions

B.Z., Y.C. and X.L. designed the experiments. B.Z., Y.C., N.J., L.Y., S.S., Y.Z., Z.W., L.R., H.L. and G.H. performed the experiments. B.Z., Y.C. and N.J. analyzed the data. B.Z. and X.L. supervised the work. B.Z., Y.C. and X.L. wrote the paper.

## Additional information

**Competing interests:** The authors declare no competing interests.

