## [Peer Review File · Nature Communications]

Reviewers' comments:

Reviewer #1 (Remarks to the Author):

In this study, the authors examine the role of *Znhit1*, a component of the SRCAP chromosome remodeling complex, within intestinal stem cells under homeostatic conditions. A new floxed *Znhit1*^{fl/fl} mouse allele is generated and indeed a substantial amount of mouse genetics is employed. Loss of *Znhit1* does not exert obvious effects on embryogenesis. Instead, the authors find *Znhit1* is required for maintaining postnatal intestinal homeostasis. The authors further propose a molecular model, indicating that ZNHIT1-mediated H2A.Z incorporation is required for intestinal stem cell specific gene *Lgr5* expression at transcriptional level. Thus, the authors conclude that *Znhit1* plays a critical role in *Lgr5*⁺ ISCs specification and maintenance.

In general, the manuscript is easy to follow, the experiments well controlled and many genetic crosses are performed. Certainly, *Znhit1* would be a novel regulator of intestinal stem cells. However, several conclusions are somewhat problematic particularly the attribution of stem cell phenotypes to general epithelial-specific deletion instead of stem cell-specific deletion, and lack of knowledge of whether *Znhit1* is expressed solely in ISCs or more generally in the epithelium.

Major comments:

1. Without knowing the expression pattern of *Znhit1* in the intestinal epithelium, it is difficult to make a strong statement on the role of *Znhit1* in *Lgr5*⁺ ISCs specification or homeostasis by comprehensively knocking out *Znhit1* in the intestinal epithelium under VillinCre (or CreERT) background. It is critical to establish the expression pattern of *Znhit1* in the intestine. Is it expressed in *Lgr5*⁺ ISC? Transit amplifying? Differentiated cells? Villi? If *Znhit1* is performing some kind of ubiquitous housekeeping function in many cell types besides stem cells then that changes the interpretation. For instance, some aspects of the growth retardation and lethality with villin-Cre (Fig 1) could arise from impairment of differentiated cell function instead of stem cell function. And apparently milder phenotypes are seen with the *Olfm4*-CreER deletion in Fig 2 although long time points do not seem to be analyzed. To address the role of *Znhit1* in *Lgr5*⁺ ISCs, it is surprising that the authors do not explore any possibilities further by using *Lgr5*-EGFP-IRES-CreERT2; *Znhit1*^{fl/fl} mice although *Olfm4*-CreER is a reasonable alternative. Stem-cell specific deletion of *Znhit1* should be employed as much as possible to derive stem-cell specific conclusions and the *Olfm4*-CreER for some reason is underutilized.

2. In addition it is well known that complete loss of the *Lgr5*⁺ ISCs will not perturb homeostasis of the epithelium. Several distinct cell populations, such as +4 *Bmi1*⁺ cells (Nature, 2011 Sep 18;478(7368):255-9.), *Dll1*⁺ secretory progenitors (Nat Cell Biol, 2012 Oct;14(10):1099-1104.), and *Alpi*⁺ enterocyte progenitors (Cell Stem Cell, 2016 Feb 4;18(2):203-13.), can compensate for the elimination of *Lgr5*⁺ ISCs. These results conflict the conclusion the authors have made in this manuscript, which is *Znhit1* plays a critical role in maintaining intestinal homeostasis through supporting *Lgr5*⁺ ISCs. In other word, the decrease of *Lgr5* expression in *Znhit1* deletion is not sufficient to explain the intestinal phenotype described in this manuscript.

3. Overall, how do the authors reconcile decreased *Lgr5*/*Olfm4*⁺ ISC with the increased proliferation in the *Znhit1* ko mice?

4. In Figure 2d and 2e. The authors specifically delete *Znhit1* in *Lgr5*⁺ ISCs by using *Olfm4*-IRES-eGFPcreERT2, and conclude *Lgr5*⁺ ISCs "stemness" is decreased in absent of *Znhit1* based on the

protein level of eGFP reporter without any other functional assays or validation of gene expression. In Fig 2d, there appears to be a decrease in the GFP fluorescence per cell and *Olfm4*-IRES-eGFP^{creERT2}; *Znhit1*^{fl/fl} mice still contain many GFP⁺ (*OLFM4*⁺) ISCs, suggesting *Olfm4* mRNA is still transcribed in the absence of *Znhit1*. In Fig 2e can the authors provide quantitation of the numbers of GFP⁺ cells as opposed to the relative expression level of GFP? These results appear to conflict with the conclusion in Figure 2b and 2c, and the model proposed later in Figure 4 and Figure 5.

5. In Figure 3e the authors state that activation of Wnt signaling and inhibition of TGF- β signaling cooperate to rescue the *Znhit1* deficient organoids in vitro. However, how can upstream signals rescue the phenotypic defects caused by downstream factors, ZNHIT1 and H2A.Z, at the transcriptional level? This data may actually demonstrate that the function of ZNHIT1, especially in maintaining *Lgr5*⁺ ISCs, may not be as critical as the authors claim.

6. In Figure 5d and 5e. I think a more thorough understanding of the mechanism underlying the *Znhit1*/H2A.Z-mediated transcription would significantly improve the paper. In *Znhit1*-deficient crypts, the phosphorylated YL1 is significantly decreased (Figure 5a and 5b). If the disruption of binding between non-phosphorylated YL1 and H2A.Z in *Znhit1*-deficient crypts (Figure 5d) is the main reason for the change of *Lgr5* and *Tgfb1* gene expression, where is the role of phosphorylated YL1 in this proposed model (Figure 5e)? Do the authors expect to see any difference of gene expression, protein-protein interaction, or H2A.Z in WT crypts (organoids) treated with phosphatase? or specifically engineering WT organoids with an YL1 mutant that mimics phosphorylated YL1 or non-phosphorylated YL1?

Minor comments:

1. In Figure 1f, 2c and 2f, the data are based on 2~3-day organoid culture where longer time points would be more convincing. Also, detailed descriptions are missing. Do the authors plate equal amounts of crypts per group? How many crypts did they plate? Quantification of survival rate and organoids growth curve (size and number) for at least 7 days will be helpful for making conclusions on stem cell activity.

2. In Figure 2b, *Olfm4* is a known Notch target in *Lgr5*⁺ ISC. Does *Znhit1* deletion affect Notch activity (NICD, HES1 ...etc.)? Since Notch activity is important for cell fate decision in the intestine. Does *Znhit1* deletion affect intestinal absorptive vs. secretory cells (*ATOH1*, *MUC2*, *CHGA* ... etc.) fate decision? Do the authors identify H2A.Z binding peaks on the *Olfm4* promoter region in their ChIP-seq (Figure 4a)?

3. In Figure 3, the authors perform RNA-seq from WT and *Znhit1* deficiency crypts. Ideally the sample number would be increased to be at least N=3 per group as it is difficult to derive statistical significance from N=1.

4. In Figure 3b, *Lgr5*, *Pcdh8*, *Olfm4* and *Clic6* are selected from 510 *Lgr5*⁺ ISC signature genes. The overall rationale for this approach is not entirely clear but even less clear is the rationale based on how many *Lgr5*⁺ ISC signature genes are significantly affected due to *Znhit1* deletion. Are there only 4 *Lgr5*⁺ ISC signature genes significantly changed in *Znhit1* deletion crypts? Although the authors refer to them as "well-characterized" *Lgr5*⁺ ISC genes, some of these genes, such as *Pcdh8* and *Clic6*, to my knowledge, has not previously been functionally characterized in *Lgr5*⁺ ISCs.

5. In Figure 3e, how many days (hours) are these organoids treated by CHIR99021 and SB431542?

6. In Figure 4a, the sequencing depth for the ChIP-seq is rather low, and performed in only one sample (N=1) per group. The authors need to prove the quality of ChIP-seq and provide detail data preprocessing and analysis. For example, how many reads have been done for each sample? What's the mappability? ...etc.. It is also not clear if all of the datasets are deposited in a repository for public access.
7. In Figure 4b are these 130 Znhit1-regulated genes enriched in published LGR5 signature gene lists? What kind of signaling pathways or biological functions are enriched in these genes? GO analysis and/or GSEA analysis could answer the question.
8. In Figure 5c the authors state "... the phosphorylated YL1 showed stronger affinity with H2A.Z ..." (Line 231). It is difficult to formally conclude this from Western blots. Without quantification or further biochemical determination of binding affinity, the authors may wish to temper these conclusions.
9. In Supplementary Figure 3b. The Lysozyme staining is hard to see. The authors state "Znhit1 deficiency led to expansion of Paneth population,...." (Line 180). However, in this image, looks like these LYZ+ cells are mislocated and not mature. Therefore, the upregulation of Lyz2 in Znhit1 deficiency mice is not necessary to be the expansion of Paneth cells.
10. In the Discussion, lines 255-257 the authors mention a crypt fission phenotype in Znhit1 deficiency mice. However, they did not show any data to support this observation in this manuscript.
11. Also in the discussion, lines 300-303. The authors state "Our findings not only provide a potential target in treatment of gastrointestinal epithelium-related diseases ...", however; this conclusion may be overstated. I don't see clear evidence that can support this sentence in this manuscript. The authors should at least explain how and/or why Znhit1/H2A.Z could be targeted.
12. Line 203. "Fig. 3b should be Fig. 4b".

Reviewer #3 (Remarks to the Author):

In the manuscript entitled 'Znhit1 controls intestinal stem cell specification by regulating H2A.Z incorporation' the authors use organoid culture, mouse genetics and ChIP-Seq to investigate the role of Znhit1. They propose that Znhit1 is essential for Lgr5+ ISC maintenance. By mediating the incorporation of the histone H2A.Z into the TSS of stemness-related gene involved in the Lgr5+ ISC fate determination, thereby promoting their expression of.

The authors provide novel information on how the gene Znhit1 affect stem cell maintenance and show that this gene is essential in postnatal and adult stem cell maintenance. In addition, it provides mechanistic clues to how Znhit1 exert its function.

If the authors can address the following concerns I believe the manuscript can be published in Nature Communications.

Major concerns

- Enlarged crypts and defective villi could also be a result of a differentiation defect. For example if the daughter cells get stuck in a progenitor state and/or fail to migrate out of the crypt. Thus, the authors should stain for progenitors as well as the differentiated cell types.

- The authors want to couple *Znhit1* KO with an overexpression of *Tgf-β* and consequently increased Paneth cell differentiation. The evidence supporting an upregulation of the *TGF-β* pathway is mainly on a transcription level. *Tgfb1* and *Tgfb2* need confirmation of their increased protein expression level with antibody staining.
 - The authors suggest that *Zhit1* regulates the expression of *Lgr5* in a Wnt independent way, based on the unaffected level of transcription of Wnt target genes such as *Axin2* and *Ascl2*. However, to rescue *Znhit1* KO organoids both Wnt hyperactivation and *TGF-β* inhibition is necessary. This indicates that Wnt signalling might be affected. TOP-FOP assays as well as in situ of Wnt target genes could be used to investigate whether the Wnt pathway is indeed unaffected.
 - Do H2A.Z KO organoids recapitulate the phenotype of the *Znhit1* KO organoids?
 - Does phosphorylation of YL1 rescue *Znhit1* KOs and/or does mutation of the phosphorylation sites of YL1 recapitulate the *Znhit1* KO phenotype in mice/organoids having normal levels of *Znhit1*?
 - The interaction between H2A.Z and YL1 has already been established. To add novelty to the mechanism the authors need to prove that *Znhit1* interacts with and phosphorylates YL1. The authors should perform IP of *Znhit1* to identify its targets.
- Minor concerns
- The lysozyme staining of the control in Fig3b is expected to be located in the bottom of the crypts. The authors should provide larger figures of high resolution so that it is easy to judge the quality of the staining.
 - The authors talk about ablation of *Lgr5+* cells. 'Ablation' is a term commonly used in correlation to induced cell death at will. 'Depletion' would have been a better word for that. Do the *Lgr5+* cells die or differentiate upon *Znhit1* KO? Caspase3 staining as well as staining for the different cell types could help on this.
 - Be consistent with the spelling of gene names e.g. Is it YL1 or YL-1.
 - There is still a faint band in the *Znhit1*/fl western. Could the authors please comment on this.
 - Professional proof-reading by a qualified or native English speaking person is needed.

Reviewer #4 (Remarks to the Author):

This manuscript examines the function of *znhit1*, a component of a chromatin remodeling complex, in intestinal homeostasis and stem cell maintenance. They demonstrate that ablation of *znhit1* in the intestinal epithelium leads to loss of the *Lgr5+* stem cells. *Znhit1* incorporates H2AZ in the promoters of genes involved in stem cell function. The authors also show that *znhit1* does so by controlling the phosphorylation of the histone H2AZ chaperone *yl-1*.

This is a very interesting manuscript, that convincingly shows the involvement of *znhit1* in stem cell maintenance (and not specification) in the intestine. However, the mechanistic data presented are rather thin and too preliminary to warrant publication in its current form.

My main concern is that not enough info is there as to how *znhit1* affects phosphorylation of *yl-1* (or whether indeed it is phosphorylation of *yl-1* that is affected). The authors should make an effort to

characterize the modification in more detail (which is the region and/or aa modified; do the consensus motifs point to putative responsible kinases; are those kinase putatively deregulated by *znhit1* ko). An interesting issue also concerns the difference in transcriptional output that H2AZ loss brings about in different gene categories (negative in *Igr5*, positive in *tgfb1*). While I do appreciate that this is a big issue and potentially beyond reasonable expectation for a single manuscript, the authors have not presented any effort towards understanding where this difference comes from (i.e. do the up- and down-regulated genes have transcription factor motifs in common in their promoter sequences – repressor or activator – that are occluded by H2AZ? Do these motifs point to signaling pathways responsible for this difference? Are these pathways deregulated in *znhit* ko?)

Another major point that needs addressing is where *znhit1* is expressed in the intestine. Is it stem cell enriched or expressed throughout? An *in situ* and/or *ihc* would help.

Minor issues to be dealt with:

1. Why do the authors not try to knock out *znhit1* in *Igr5* cells directly (using an *Igr5-cre*)? *Olfm4-cre* is a suitable substitute, but still only a substitute.
2. An *Igr5 in situ* would also be useful to show the extent of its ablation upon *znhit1* ko.
3. In Figure 1g the genotypes are I think reversed.
4. In Figure 2 a quantitation of *Igr5* levels after *znhit1* ablation in *olfm4* cells would be suitable.
5. Figure 2f should be quantitated, as the effect is not particularly pronounced. Could that be due to escaper crypts that have not recombined *znhit1*?
6. In figure 3e, what happens to *Igr5* expression after the treatments? What happens to other *znhit1* ko affected genes, both up- and down-regulated ones?
7. In figure 4b, there should be a p-value attached, as to how significant the observed overlap is.
8. Are the NGS data deposited somewhere (GEO or other)?
9. The English could do with some polishing. I present just a couple of examples (there are many more, not terribly serious mistakes, in the text, that do need attention):
 - a. Line 133: which led to following body weight decrease and intestinal epithelium transformation
Should be: which led to concomitant body weight decrease and intestinal epithelium degeneration (or some such)
 - b. Line 239: play dominate roles
Should be: plays a dominant role (or some such).

Figure changes

Revised Figure	Previous Figure	Modification
Fig. 1h	Fig. 1h	New data added
Fig. 2d	Fig. 2d	Quantification added
Fig. 2f	Fig. 2f	Replaced with new data
Fig. 3a	Fig. 3a	Biological replicates added
Fig. 3b	Fig. 3b	Replaced with new data
Fig. 3c		New data
Fig. 3d	Fig. 3c	
Fig. 3e	Fig. 3d	
Fig. 3f	Fig. 3e	
Fig. 3g		New data
Fig. 4b	Fig. 4b	Replaced with new data
Fig. 4f		New data
Fig. 5b		New data
Fig. 5c	Fig. 5b	
Fig. 5d	Fig. 5c	Quantification added
Fig. 5e		New data
Fig. 5f		New data
Fig. 5g	Fig. 5d	
Fig. 5h	Fig. 5e	
Supplementary Fig. 1		New data
Supplementary Fig. 2	Supplementary Fig. 1	
Supplementary Fig. 3	Supplementary Fig. 2	
Supplementary Fig. 4		New data
Supplementary Fig. 5a	Supplementary Fig. 3a	
Supplementary Fig. 5b		New data
Supplementary Fig. 6	Supplementary Fig. 3b	Replaced with new data
Supplementary Fig. 7		New data
Supplementary Fig. 8a,b	Supplementary Fig. 4a,b	
Supplementary Fig. 8c		New data
Supplementary Fig. 9	Supplementary Fig. 5	New data added
Supplementary Tab. 1	Supplementary Tab. 1	Replaced with new data

REVIEWER 1:

In this study, the authors examine the role of *Znhit1*, a component of the SRCAP chromosome remodeling complex, within intestinal stem cells under homeostatic conditions. A new floxed *Znhit1*^{fl/fl} mouse allele is generated and indeed a substantial amount of mouse genetics is employed. Loss of *Znhit1* does not exert obvious effects on embryogenesis. Instead, the authors find *Znhit1* is required for maintaining postnatal intestinal homeostasis. The authors further propose a molecular model, indicating that ZNHIT1-mediated H2A.Z incorporation is required for intestinal stem cell specific gene *Lgr5* expression at transcriptional level. Thus, the authors conclude that *Znhit1* plays a critical role in *Lgr5*⁺ ISCs specification and maintenance.

In general, the manuscript is easy to follow, the experiments well controlled and many genetic crosses are performed. Certainly, *Znhit1* would be a novel regulator of intestinal stem cells. However, several conclusions are somewhat problematic particularly the attribution of stem cell phenotypes to general epithelial-specific deletion instead of stem cell-specific deletion, and lack of knowledge of whether *Znhit1* is expressed solely in ISCs or more generally in the epithelium.

Major comments:

1. Without knowing the expression pattern of *Znhit1* in the intestinal epithelium, it is difficult to make a strong statement on the role of *Znhit1* in *Lgr5*⁺ ISCs specification or homeostasis by comprehensively knocking out *Znhit1* in the intestinal epithelium under VillinCre (or CreERT) background. It is critical to establish the expression pattern of *Znhit1* in the intestine. Is it expressed in *Lgr5*⁺ ISC? Transit amplifying? Differentiated cells? Villi?

Response:

Thank the reviewer for the constructive comments. Indeed, it is critical to address the expression pattern of *Znhit1* in the intestine. As the available anti-*Znhit1* antibody cannot give specific nuclear staining, we examine *Znhit1* expression level in different epithelial parts (villi and crypts) and particular cell types (*Lgr5*⁺ ISCs, daughter progenitor cells and other crypt cells) employing precise isolation followed by RT-qPCR.

First, we mechanically isolate intestinal villi and crypts from 8-week-old C57BL/6 mice. New Supplementary Fig. 1a shows that *Znhit1* is mainly expressed in intestinal crypts. Then, we dissociate *Lgr5-EGFP-IRE5-creERT2* crypts into single cells and sort *Lgr5*⁺ ISCs (GFP^{hi}), daughter progenitor cells (GFP^{low}) and other crypt cells (GFP^{neg}) using FACS (new Supplementary Fig. 1b). We find that *Lgr5*⁺ ISCs have robust *Znhit1* expression, while their daughter progenitor cells and other crypt cells have significantly reduced *Znhit1* expression (new Supplementary Fig. 1b). This

ISC-enriched expression pattern of *Znhit1* supports its primary function in determining the fate of *Lgr5*⁺ ISCs.

If *Znhit1* is performing some kind of ubiquitous housekeeping function in many cell types besides stem cells then that changes the interpretation. For instance, some aspects of the growth retardation and lethality with villin-Cre (Fig 1) could arise from impairment of differentiated cell function instead of stem cell function.

Response:

To address the reviewer's concern that *Znhit1* might exert housekeeping function in many cell types besides stem cells, we perform immunostaining to reveal the TA cells and differentiated cells in intestinal epithelium. We find that *Znhit1*-deficient mice show comparable presence of TA cells (marked by Ki67-Fig. 1e), enterocytes (pan-differentiation marked by Krt20-Fig. 1e), goblet cells (marked by Mucin2-the following Attached Fig. 1), enteroendocrine cells (marked by Chr-A-the following Attached Fig. 2) and Paneth cells (marked by Lysosome-new Supplementary Fig. 6).

Attached Figure 1 | *Znhit1* deletion has no obvious effect on the differentiation of goblet cells. Scale bar, 50 μ m.

Attached Figure 2 | *Znhit1* deletion has no obvious effect on the differentiation of enteroendocrine cells. Scale bar, 50 μ m.

And apparently milder phenotypes are seen with the *Olfm4*-CreER deletion in Fig 2 although long time points do not seem to be analyzed.

Response:

As the reviewer pointed out, the milder phenotypes of *Znhit1^{fl/fl}*; *Olfm4-IRES-eGFPcreERT2* mice were due to short-term *Znhit1* knockout (3-day tamoxifen treatment followed by 4-day waiting period, not 4-day tamoxifen treatment followed by 7-day waiting period). We carefully designed this time point to demonstrate that *Znhit1* determines the fate of *Lgr5*⁺ ISC in a cell autonomous manner. As *Lgr5*⁺ ISCs continuously generate all cell types in intestinal epithelium during homeostasis, longer time point might not exclude the contribution of other cell types to *Lgr5*⁺ ISCs depletion through changing the niche.

To address the reviewer's concern, we apply the "4+7" strategy to *Znhit1^{fl/fl}*; *Olfm4-IRES-eGFPcreERT2* mice and observe significant body weight decrease after day 7 (the following Attached Fig. 3). This indicates that *Znhit1* loss in *Lgr5*⁺ ISCs does lead to epithelium defects.

Attached Figure 3 | *Znhit1* loss in *Lgr5*⁺ ISCs leads to significant body weight decrease. Eight-week-old *Znhit1^{+/+}*; *Olfm4-IRES-eGFPcreERT2* and *Znhit1^{fl/fl}*; *Olfm4-IRES-eGFPcreERT2* mice were daily injected with tamoxifen for 4 days followed by 7-day waiting period. Top: Scheme of Cre induction strategy. Bottom: Body weight comparison between *Znhit1^{+/+}*; *Olfm4-IRES-eGFPcreERT2* and *Znhit1^{fl/fl}*; *Olfm4-IRES-eGFPcreERT2* mice at indicated time following tamoxifen treatment. (n=3).

To address the role of *Znhit1* in *Lgr5*⁺ ISCs, it is surprising that the authors do not explore any possibilities further by using *Lgr5-EGFP-IRES-CreERT2*; *Znhit1^{fl/fl}* mice although *Olfm4*-CreER is a reasonable alternative. Stem-cell specific deletion of *Znhit1* should be employed as much as possible to derive stem-cell specific conclusions and the *Olfm4*-CreER for some reason is underutilized.

Response:

Although *Lgr5-EGFP-IRES-creERT2* strain is widely employed to characterize Lgr5+ ISCs, the expression of creERT2 is silenced in patches of crypts¹, which makes the strain inappropriate for investigating the contribution of Lgr5+ ISCs negative regulation to epithelial homeostasis (rapid compensation from adjacent crypts).

To our knowledge, *Olfm4-IRES-eGFPcreERT2* is the best tool to delete gene specifically in Lgr5+ ISCs throughout the intestinal epithelium with ISCs GFP-marked. Of note, we confirmed key results in *Znhit1^{fl/fl}*; *Olfm4-IRES-eGFPcreERT2* mice, including depletion of Lgr5+ ISCs (*in vivo*-Fig. 2d,e and *in vitro*-Fig. 2f and Fig. 3f) and changed expression of critical *Znhit1/H2A.Z* target genes (Fig. 3d and new Fig. 3g).

2. In addition it is well known that complete loss of the Lgr5+ ISCs will not perturb homeostasis of the epithelium. Several distinct cell populations, such as +4 *Bmi1*+ cells (Nature, 2011 Sep 18;478(7368):255-9.), *Dll1*+ secretory progenitors (Nat Cell Biol, 2012 Oct;14(10):1099-1104.), and *Alpi*+ enterocyte progenitors (Cell Stem Cell, 2016 Feb 4;18(2):203-13.), can compensate for the elimination of Lgr5+ ISCs. These results conflict the conclusion the authors have made in this manuscript, which is *Znhit1* plays a critical role in maintaining intestinal homeostasis through supporting Lgr5+ ISCs. In other word, the decrease of Lgr5 expression in *Znhit1* deletion is not sufficient to explain the intestinal phenotype described in this manuscript.

Response:

As the reviewer pointed out, we did consider that “+4” ISCs might compensate the role of Lgr5+ ISCs in maintaining intestinal homeostasis. However, deleting *Znhit1* in intestinal epithelium, either using *Villin-cre* or *Villin-creERT*, does not affect the expression of “+4” ISC marker genes (*Bmi1*, *Hopx* and *Lrig1*²) in intestinal crypts (the following Attached Fig. 4). This indicates that the “+4” ISC population is present but cannot regenerate intestinal epithelium (*in vivo*) (Fig. 1a-e and Fig. 2a) or organoids (*in vitro*) (Fig. 1f and Fig. 2c). In addition, Lgr5+ ISC-specific *Znhit1* deletion led to the failure of organoid generation (new Fig. 2f), which could not be compensated by wild-type “+4” ISCs. One possibility is that *Znhit1* preserves the stemness of Lgr5+ ISCs meanwhile restricts their secretion of *Tgfb1* (new Fig. 3c), which is essential for crypt niche maintenance. The interaction between Lgr5+ ISCs and “+4” ISCs through crypt niche is still unclear and requires further studies.

Attached Figure 4 | Znhit1 deficiency depletes Lgr5+ ISCs but not “+4” ISCs in intestinal crypts. (a) Intestinal crypts were harvested from *Znhit1*^{fl/+}; *Villin-cre* (fl/+) and *Znhit1*^{fl/fl}; *Villin-cre* (fl/fl) mice at P9 for qRT-PCR to examine the expression of indicated genes. (b) Eight-week-old *Villin-creERT* (+/+) and *Znhit1*^{fl/fl}; *Villin-creERT* (fl/fl) mice were daily injected with tamoxifen for 4 days followed by 7-day waiting period. Intestinal crypts were harvested for qRT-PCR to examine the expression of indicated genes. Histone H3 was used as an internal control. The statistical data represent mean±s.d. (n=3).

3. Overall, how do the authors reconcile decreased Lgr5/Olfm4+ ISC with the increased proliferation in the *Znhit1* ko mice?

Response:

In homeostasis condition, Ki67 mainly marks fast-proliferating TA cells³. As described in the manuscript, *Znhit1* deficiency leads to enlarged crypt zone, in which the Lgr5+ ISCs are depleted (Fig. 1d-h). Therefore, the expansion of TA cells in the enlarged crypt zone might be the reason of increased Ki67 staining.

4. In Figure 2d and 2e. The authors specifically delete *Znhit1* in Lgr5+ ISCs by using *Olfm4-IRES-eGFPcreERT2*, and conclude Lgr5+ ISCs “stemness” is decreased in absent of *Znhit1* based on the protein level of eGFP reporter without any other functional assays or validation of gene expression.

Response:

For *Lgr5*⁺ ISC functional assay, we examine *in vitro* organoid generating ability in new Fig. 2f (7-day culture shown and quantification of organoid buddings along the time added). Besides, validation of the expression of *Lgr5*⁺ ISC signature genes was provided in Fig. 3d and new Fig. 3g.

In Fig 2d, there appears to be a decrease in the GFP fluorescence per cell and *Olfm4*-IRES-eGFPcreERT2; *Znhit1*^{fl/fl} mice still contain many GFP⁺ (OLFM4⁺) ISCs, suggesting *Olfm4* mRNA is still transcribed in the absent of *Znhit1*. In Fig 2e can the authors provide quantitation of the numbers of GFP⁺ cells as opposed to the relative expression level of GFP? These results appear to conflict with the conclusion in Figure 2b and 2c, and the model proposed later in Figure 4 and Figure 5.

Response:

We add the quantitation of GFP⁺ cells in Fig. 2d, which verifies that *Znhit1* deletion leads to a significant reduce of *Lgr5*⁺ ISC number. We did observe dramatically decreased but still existing *Olfm4* transcription after short-term *Lgr5*⁺ ISC-specific *Znhit1* deletion (Fig. 3d). However, this does not conflict with our model. *Olfm4*-GFP fluorescence decrease (Fig. 2d), *Olfm4*-GFP protein level decrease (Fig. 2e) and *Olfm4* mRNA level decrease (Fig. 3d) all demonstrated that *Znhit1* sustains *Olfm4* expression in *Lgr5*⁺ ISCs.

5. In Figure 3e the authors state that activation of Wnt signaling and inhibition of TGF- β signaling cooperate to rescue the *Znhit1* deficient organoids *in vitro*. However, how can upstream signals rescue the phenotypic defects caused by downstream factors, ZNHIT1 and H2A.Z, at the transcriptional level? This data may actually demonstrate that the function of ZNHIT1, especially in maintaining *Lgr5*⁺ ISCs, may not as critical as the authors claim.

Response:

Although *Znhit1* and H2A.Z are transcriptional regulators, they are not at downstream of Wnt or TGF- β signalling.

The Wnt signalling activity, which is indicated by the transcription level of classic Wnt target gene *Axin2*, is unaffected in *Znhit1*-deficient crypts (qPCR in Supplementary Fig. 5a and *in situ* in new Supplementary Fig. 5b). These data suggest that *Znhit1* and Wnt signalling control *Lgr5* expression in parallel. We employed CHIR99021 to rescue the phenotypic defect because Wnt hyperactivation could sustain the expression of *Lgr5* in *Znhit1*-deficient organoid (new Fig. 3g).

For TGF- β signalling, *Znhit1* and H2A.Z directly suppress the transcription of TGF- β ligand (*Tgfb1*) and receptor (*Tgfb2*) (Fig. 3a-d and Fig. 4c-f) thus restrict TGF- β

signalling (Fig. 3e), indicating that *Znhit1* and *H2A.Z* work at the upstream of TGF- β signalling.

6. In Figure 5d and 5e. I think a more thorough understanding of the mechanism underlying the *Znhit1*/*H2A.Z*-mediated transcription would significantly improve the paper. In *Znhit1*-deficient crypts, the phosphorylated YL1 is significantly decreased (Figure 5a and 5b). If the disruption of binding between non-phosphorylated YL1 and *H2A.Z* in *Znhit1*-deficient crypts (Figure 5d) is the main reason for the change of *Lgr5* and *Tgfb1* gene expression, where is the role of phosphorylated YL1 in this proposed model (Figure 5e)? Do the authors expect to see any difference of gene expression, protein-protein interaction, or *H2A.Z* in WT crypts (organoids) treated with phosphatase? or specifically engineering WT organoids with an YL1 mutants that mimic phosphorylated YL1 or non-phosphorylated YL1?

Response:

We appreciate the reviewer's suggestion that providing more detailed mechanism will significantly benefit the manuscript.

To address the reviewer's concern "the disruption of binding between non-phosphorylated YL1 and *H2A.Z* in *Znhit1*-deficient crypts (Fig. 5d) is the main reason for the change of *Lgr5* and *Tgfb1* gene expression", we demonstrate that dephosphorylating YL1 by shrimp alkaline phosphatase eliminates the binding of YL1 to *H2A.Z* (new Fig. 5e). Besides, we address p-Akt as a YL1 kinase (new Fig. 5b,f). We find that inhibiting Akt activity ablates YL1 phosphorylation thus abolishes the interaction between YL1 and *H2A.Z* (new Fig. 5f).

Honestly, it is difficult to manipulate YL1 phosphorylation using shrimp alkaline phosphatase upon live organoid. We have tried but not yet identified the YL1 phosphorylation site.

Minor comments:

1. In Figure 1f, 2c and 2f, the data are based on 2~3-day organoid culture where longer time points would be more convincing. Also, detailed descriptions are missing. Do the authors plate equal amounts of crypts per group? How many crypts did they plate? Quantification of survival rate and organoids growth curve (size and number) for at least 7 days will be helpful for making conclusions on stem cell activity.

Response:

As quantitated in Fig. 1f and 2c, rare *Znhit1*-deficient crypt could generate organoid following dissociation. We provide the pictures taken at day 5 in the following Attached Fig. 5 for reviewer's inquiry.

Attached Figure 5 | *Znhit1*-deficient intestinal crypts loss the ability to generate organoids *in vitro*. Intestinal crypts were isolated from *Znhit1^{fl/+}; Villin-cre* and *Znhit1^{fl/fl}; Villin-cre* mice at P9, embedded in Matrigel (100 crypts per well) and cultured for 5 days.

In new Fig. 2f, we replace 3-day culture with 7-day culture and provide quantification of organoid buddings along the time for better illustration.

We did plate equal number of crypts (100 crypts each well). This detail is added in figure legends.

2. In Figure 2b, *Olfm4* is a known Notch target in *Lgr5+* ISC. Does *Znhit1* deletion affect Notch activity (NICD, HES1 ...etc.)? Since Notch activity is important for cell fate decision in the intestine. Does *Znhit1* deletion affect intestinal absorptive vs. secretory cells (*ATOH1*, *MUC2*, *CHGA* ... etc.) fate decision? Do the authors identify H2A.Z binding peaks on the *Olfm4* promoter region in their ChIP-seq (Figure 4a)?

Response:

We examine NICD protein level and find that *Znhit1* deletion has no obvious effect on Notch activity (the following Attached Fig. 6).

Attached Figure 6 | Znhit1 deficiency has no obvious effect on Notch signalling activity. Eight-week-old *Villin-creERT* (+/+) and *Znhit1^{fl/fl}*; *Villin-creERT* (fl/fl) mice were daily injected with tamoxifen for 4 days followed by 7-day waiting period. Intestinal crypts were harvested for immunoblotting with the indicated antibodies. GAPDH served as a loading control.

We did not identify any significant H2A.Z peak on *Olfm4* TSS region in ChIP-seq.

3. In Figure 3, the authors perform RNA-seq from WT and *Znhit1* deficiency crypts. Ideally the sample number would be increased to be least N=3 per group as it is difficult to derive statistical significance from N=1.

Response:

We increase the RNA-seq data to 3 biological replicates for wild-type and *Znhit1*-deficient crypts respectively (new Fig. 3a). The principal component analysis (PCA) clearly separates the 6 sequenced samples into 2 groups, which are consistent with the genotypes. Then, these RNA-seq datasets are used for statistical test analysis.

Attached Figure 7 | Principle component analysis (PCA) of the gene expression profiles across all 6 sequenced samples.

4. In Figure 3b, *Lgr5*, *Pcdh8*, *Olfm4* and *Clic6* are selected from 510 *Lgr5*⁺ ISCs signature genes. The overall rationale for this approach is not entirely clear but even less clear is the rationale based on how many *Lgr5*⁺ ISCs signature genes are significantly affected due to *Znhit1* deletion. Are there only 4 *Lgr5*⁺ ISCs signature genes significantly changed in *Znhit1* deletion crypts? Although the authors refer to them as “well-characterized” *Lgr5*⁺ ISCs genes, some of these genes, such as *Pcdh8* and *Clic6*, to my knowledge, has not previously functionally characterized in *Lgr5*⁺ ISCs.

Response:

As the analysis was performed at the transcription level, we focused on the 384 Lgr5+ISCs mRNA signature genes⁴. In the revised manuscript, RNA-seq data are increased to 3 biological replicates according to the reviewer's suggestion. Across 172 Znhit1-downregulated genes, 15 genes have been identified as Lgr5+ ISCs mRNA signature genes (new Supplementary Fig. 4). The fisher's exact test indicates the significant enrichment of Znhit1-downregulated genes in Lgr5+ISCs signature group with P-value= 2.73×10^{-10} .

Pcdh8 was verified as a Lgr5+ ISC signature gene by *in situ* assay⁵. *Clic6* was addressed as a Lgr5+ ISC signature gene by both Affymetrix and Agilent⁴. As new RNA-seq results exclude *Pcdh8*, we verify more Znhit1-downregulated Affymetrix and Agilent signature genes (*Dach1*, *Essrg* and *Scn2b*) using RT-qPCR in Fig. 3b.

5. In Figure 3e, how many days (hours) are these organoids treated by CHIR99021 and SB431542?

Response:

The treatment was present from crypt embedding to final examination (4 days).

6. In Figure 4a, the sequencing depth for the ChIP-seq is rather low, and performed in only one sample (N=1) per group. The authors need to prove the quality of ChIP-seq and provide detail data preprocessing and analysis. For example, how many reads have been done for each sample? What's the mappability? ...etc.. It is also not clear if all of the datasets are deposited in a repository for public access.

Response:

For each sample, the Chip-seq was sequenced by Illumina Hiseq2500 with 1×50bps. Sequenced reads were aligned to reference genome (mm10) using Bowtie (v1.1.1). The statistical performances of each Chip-seq data were summarized in the following Attached Table 1. Around 33~42M (millions) reads were sequenced for each sample, respectively. About 97.9%~98.7% of sequenced reads could be mapped to reference genome (mm10) with no more than 2 mismatches. The distribution of uniquely mapped reads across different genomic regions was summarized in the following Attached Table 2. There were about 10~30 reads uniquely mapped to different genomic regions per 1 kb length. And we found the reads were relatively enriched in the 5'UTR-exon and TSS_up_1kb regions with Fold Change (FC) =2.23 and 1.55. Peaks calling analysis was performed by MACS with default cutoff.

Attached Table 1 | Statistical summary of Chip-seq alignment analysis.

Sample ID	Number of reads	Number of mapped reads	Mapped ratio (%)	Uniquely mapped ratio (%)	Multiple mapped	Uniquely mapped
-----------	-----------------	------------------------	------------------	---------------------------	-----------------	-----------------

f1/f1-Input	41,580,357	41,030,413	98.68%	77.22%	8,922,709	32,107,704
f1/f1	42,030,555	41,161,353	97.93%	78.67%	8,096,759	33,064,594
+/+ -Input	32,653,997	32,228,052	98.70%	76.76%	7,164,457	25,063,595
+/+	40,539,677	39,765,506	98.09%	79.45%	7,557,520	32,207,986

Attached Table 2 | Distribution of uniquely mapped reads across different genomic regions.

Sample	Number of uniquely mapped reads per 1 Kb							
	ID	5'UTR_Exons	3'UTR_Exons	Introns	TSS_up_1kb	TSS_up_5kb	TES_down_1kb	TES_down_5kb
f1/f1-Input		16.24	15.78	16.29	11.04	10.31	14.69	11.49
f1/f1		19.3	16.57	16.54	11.37	10.61	14.38	11.66
+/+ -Input		12.05	12.15	12.72	8.65	8.04	11.35	8.86
+/+		28.2	15.97	15.96	13.44	10.97	13.63	11.13

The raw NGS data were deposited to the NCBI SRA database under accession number SRP148616 (RNA-seq data) and SRP148519 (ChIP-seq data). The data will be released upon publication.

7. In Figure 4b are these 130 *Znhit1*-regulated genes enriched in published LGR5 signature gene lists? What kind of signaling pathways or biological functions are enriched in these genes? GO analysis and/or GSEA analysis could answer the question.

Response:

New RNA-seq data switch the previous 130 *Znhit1*-regulated genes with TSS H2A.Z binding to 107 genes (new Fig. 4b and gene list in new Supplementary Tab. 1). In these 107 genes, we notice the presence of *Lgr5*+ ISC signature genes *Lgr5*, *Clic6*, *Esrrg*, *Ppp1r9a* and *Slc27a2*, which are all negatively regulated by *Znhit1* deficiency. GO analysis indicates the enrichment of “regulation of cell proliferation”, “pathway-restricted SMAD protein phosphorylation” and “organ regeneration” (new Supplementary Fig. 7).

8. In Figure 5c the authors state “... the phosphorylated YL1 showed stronger affinity with H2A.Z ...” (Line 231). It is difficult to formally conclude this from Western blots. Without quantification or further biochemical determination of binding affinity, the authors may wish to temper these conclusions.

Response:

Thanks for the suggestion. We add quantification of H2A.Z-bound p-YL1 and non-p-YL1, which indicates that phosphorylated YL1 has much stronger affinity with H2A.Z.

9. In Supplementary Figure 3b. The Lysozyme staining is hard to see. The authors state “Znhit1 deficiency led to expansion of Paneth population,....” (Line 180). However, in this image, looks like these LYZ+ cells are mislocated and not mature. Therefore, the upregulation of Lyz2 in Znhit1 deficiency mice is not necessary to be the expansion of Paneth cells.

Response:

We replace this figure with high-quality images for better illustration (new Supplementary Fig. 6). The enlargement and quantitation are added as well. As shown, *Znhit1^{fl/fl}*; *Villin-cre* mice have increased and more mature Paneth cells at P15 compared to *Znhit1^{fl/+}*; *Villin-cre* mice. This indicates that Znhit1 deficiency promotes the differentiation of Paneth cells.

10. In the Discussion, lines 255-257 the authors mention a crypt fission phenotype in Znhit1 deficiency mice. However, they did not show any data to support this observation in this manuscript.

Response:

We quantitate inter-villi regions at E18.5 and crypts at P9. As shown in the following Attached Fig. 8, Znhit1 deficiency leads to decreased crypt number at P9, suggesting impaired crypt fission after birth.

Attached Figure 8 | Quantitation of inter-villi regions at E18.5 and crypts at P9.

11. Also in the discussion, lines 300-303. The authors state “Our findings not only provide a potential target in treatment of gastrointestinal epithelium-related diseases

...”, however; this conclusion may be overstated. I don’t see clear evidence that can support this sentence in this manuscript. The authors should at least explain how and/or why *Znhit1/H2A.Z* could be targeted.

Response:

Thanks for the suggestion. We modify the sentence to avoid any over-statement. Indeed, our work emphasizes a critical physiological mechanism underlying intestinal epithelium homeostasis. The potential functions of *Znhit1/H2A.Z* in related diseases require further investigation.

12. Line 203. “Fig. 3b should be Fig. 4b”.

Response:

Thanks for the careful reading. This error is corrected.

REVIEWER 3:

In the manuscript entitled ‘Znhit1 controls intestinal stem cell specification by regulating H2A.Z incorporation’ the authors use organoid culture, mouse genetics and ChIP-Seq to investigate the role of Znhit1. They propose that Znhit1 is essential for Lgr5+ ISC maintenance. By mediating the incorporation of the histone H2A.Z into the TSS of stemness-related gene involved in the Lgr5+ ISC fate determination, thereby promoting their expression of.

The authors provide novel information on how the gene Znhit1 affect stem cell maintenance and show that this gene is essential in postnatal and adult stem cell maintenance. In addition, it provides mechanistic clues to how Znhit1 exert its function.

If the authors can address the following concerns I believe the manuscript can be published in Nature Communications.

Major concerns

- Enlarged crypts and defective villi could also be a result of a differentiation defect. For example if the daughter cells get stuck in a progenitor state and/or fail to migrate out of the crypt. Thus, the authors should stain for progenitors as well as the differentiated cell types.

Response:

Thank the reviewer for the constructive suggestion. By performing immunostaining, we find that Znhit1-deficient mice show comparable presence of progenitor cells (marked by Ki67-Fig. 1e), enterocytes (pan-differentiation marked by Krt20-Fig. 1e), goblet cells (marked by Mucin2-the following Attached Fig. 1), enteroendocrine cells (marked by Chr-A-the following Attached Fig. 2) and Paneth cells (marked by Lysosome-new Supplementary Fig. 6).

Attached Figure 1 | Znhit1 deletion has no obvious effect on the differentiation of goblet cells. Scale bar, 50 μ m.

Attached Figure 2 | *Znhit1* deletion has no obvious effect on the differentiation of enteroendocrine cells. Scale bar, 50 μ m.

- The authors want to couple *Znhit1* KO with an overexpression of Tgf- β and consequently increased Paneth cell differentiation. The evidence supporting an upregulation of the TGF- β pathway is mainly on a transcription level. *Tgfb1* and *Tgfb2* need confirmation of their increased protein expression level with antibody staining.

Response:

We perform anti-*Tgfb1* and anti-*Tgfb2* IHCs according to the reviewer's suggestion. New Fig. 3c shows that *Znhit1* deficiency indeed leads to increased protein level of both *Tgfb1* and *Tgfb2* in the intestinal crypts.

- The authors suggest that *Znhit1* regulates the expression of *Lgr5* in a Wnt independent way, based on the unaffected level of transcription of Wnt target genes such as *Axin2* and *Ascl2*. However, to rescue *Znhit1* KO organoids both Wnt hyperactivation and TGF- β inhibition is necessary. This indicates that Wnt signalling might be affected. TOP-FOP assays as well as *in situ* of Wnt target genes could be used to investigate whether the Wnt pathway is indeed unaffected.

Response:

We add *Axin2* (classic Wnt target gene) RNA scope (quantitative *in situ*) to further support that Wnt activity is unaffected in *Znhit1*-deficient crypts (new Supplementary Fig. 5b). Actually, we propose *Znhit1* and Wnt signalling control *Lgr5* expression in parallel. We employed CHIR99021 to rescue the phenotypic defect because Wnt hyperactivation could sustain the expression of *Lgr5* in *Znhit1*-deficient organoid (new Fig. 3g).

- Do H2A.Z KO organoids recapitulate the phenotype of the *Znhit1* KO organoids?

Response:

H2A.Z-deficient organoids do recapitulate the phenotype of *Znhit1*-deficient organoids. We add the data in new Supplementary Fig. 8c.

- Does phosphorylation of YL1 rescue *Znhit1* KOs and/or does mutation of the phosphorylation sites of YL1 recapitulate the *Znhit1* KO phenotype in mice/organoids having normal levels of *Znhit1*?

Response:

We appreciate the reviewer's suggestion. Honestly, we tried but not yet identified the YL1 phosphorylation site. During this revision, we demonstrate that dephosphorylating YL1 by shrimp alkaline phosphatase eliminates the binding of YL1 to H2A.Z (new Fig. 5e). Besides, we address p-Akt as a YL1 kinase (new Fig. 5b,f). We find that inhibiting Akt activity ablates YL1 phosphorylation thus abolishes the interaction between YL1 and H2A.Z (new Fig. 5f). However, as p-Akt regulates multiple pathways critical for intestinal homeostasis, examination of mice/organoid phenotype in presence of PI3K-Akt inhibitor LY294002 would not help to address the contribution of YL1.

- The interaction between H2A.Z and YL1 has already been established. To add novelty to the mechanism the authors need to prove that *Znhit1* interacts with and phosphorylates YL1. The authors should perform IP of *Znhit1* to identify its targets.

Response:

Thanks for the suggestion. We perform IP and demonstrate that *Znhit1* does interact with YL1 (new Fig. 5e).

Minor concerns

- The lysozyme staining of the control in Fig3b is expected to be located in the bottom of the crypts. The authors should provide larger figures of high resolution so that it is easy to judge the quality of the staining.

Response:

We replace this figure with high-quality images for better illustration (new Supplementary Fig. 6). The enlargement and quantitation are added as well. As shown, *Znhit1*^{fl/fl}; *Villin-cre* mice have increased and more mature Paneth cells at P15 compared to *Znhit1*^{fl/+}; *Villin-cre* mice. This indicates that *Znhit1* deficiency promotes the differentiation of Paneth cells.

- The authors talk about ablation of Lgr5+ cells. 'Ablation' is a term commonly used in correlation to induced cell death at will. 'Depletion' would have been a better word

for that. Do the Lgr5+ cells die or differentiate upon *Znhit1* KO? Caspase3 staining as well as staining for the different cell types could help on this.

Response:

Thanks for the suggestion. We change the word to “depletion” for better understanding. Indeed, no significant cell death is observed in *Znhit1*-deficient crypts revealed by anti-cleaved caspase 3 staining (the following Attached Fig. 3).

Attached Figure 3 | No significant cell death is observed in *Znhit1*-deficient crypts. Scale bar, 50 μ m.

- Be consistent with the spelling of gene names e.g. Is it YL1 or YL-1.

Response:

Thanks for the careful reading. We correct “YL-1”to “YL1” for unification.

- There is still a faint band in the *Znhit1*^{fl/fl} western. Could the authors please comment on this.

Response:

We do notice residual *Znhit1* protein expression in intestinal crypts, which is consistent with *Znhit1* mRNA level revealed by RT-qPCR (Figure 3d). This might be due to escaper crypts that have not recombined *Znhit1*.

- Professional proof-reading by a qualified or native English speaking person is needed.

Response:

We review the manuscript and correct several errors/typos. Thanks for the suggestion.

Reviewer #4 (Remarks to the Author):

This manuscript examines the function of *znhit1*, a component of a chromatin remodeling complex, in intestinal homeostasis and stem cell maintenance. They demonstrate that ablation of *znhit1* in the intestinal epithelium leads to loss of the *lgr5*⁺ stem cells. *Znhit1* incorporates H2AZ in the promoters of genes involved in stem cell function. The authors also show that *znhit1* does so by controlling the phosphorylation of the histone H2AZ chaperone *yl-1*.

This is a very interesting manuscript, that convincingly shows the involvement of *znhit1* in stem cell maintenance (and not specification) in the intestine. However, the mechanistic data presented are rather thin and too preliminary to warrant publication in its current form.

My main concern is that not enough info is there as to how *znhit1* affects phosphorylation of *yl-1* (or whether indeed it is phosphorylation of *yl-1* that is affected). The authors should make an effort to characterize the modification in more detail (which is the region and/or aa modified; do the consensus motifs point to putative responsible kinases; are those kinase putatively deregulated by *znhit1* ko).

Response:

We appreciate the reviewer's suggestion. Honestly, we tried but not yet identified the YL1 phosphorylation site. During this revision, we demonstrate that dephosphorylating YL1 by shrimp alkaline phosphatase eliminates the binding of YL1 to H2A.Z (new Fig. 5e). Besides, we address p-Akt as a YL1 kinase (new Fig. 5b,f). We find that inhibiting Akt activity ablates YL1 phosphorylation thus abolishes the interaction between YL1 and H2A.Z (new Fig. 5f).

An interesting issue also concerns the difference in transcriptional output that H2AZ loss brings about in different gene categories (negative in *lgr5*, positive in *tgfb1*). While I do appreciate that this is a big issue and potentially beyond reasonable expectation for a single manuscript, the authors have not presented any effort towards understanding where this difference comes from (i.e. do the up- and down-regulated genes have transcription factor motifs in common in their promoter sequences – repressor or activator – that are occluded by H2AZ? Do these motifs point to signaling pathways responsible for this difference? Are these pathways deregulated in *znhit* ko?)

Response:

Thank the reviewer for the constructive comment. To investigate how *Znhit1*/H2A.Z deficiency exerts opposite regulatory effects on transcription of different genes (upregulation of *Lgr5* and *Clic6*, while downregulation of *Tgfb1* and *Tgfb2*), we perform ChIP-qPCR to examine the epigenetic modification landmarks at *Lgr5* and *Tgfb1* loci in wild-type and H2A.Z-deficient crypts. As shown in new Fig. 4f, *Lgr5* TSS region has an enrichment of H3K4me3 (transcription activation landmark), while

Tgfb1 TSS region has an enrichment of H3K27me3 (transcription suppression landmark), both of which are ablated after H2A.Z deletion. These data suggest that basal epigenetic modification status might determine the regulatory effect of H2A.Z incorporation on gene transcription.

Another major point that needs addressing is where *znhit1* is expressed in the intestine. Is it stem cell enriched or expressed throughout? An in situ and/or ihc would help.

Response:

It is critical to address the expression pattern of *Znhit1* in the intestine. As the available anti-*Znhit1* antibody cannot give specific nuclear staining, we examine *Znhit1* expression level in different epithelial parts (villi and crypts) and particular cell types (*Lgr5*⁺ ISCs, daughter progenitor cells and other crypt cells) employing precise isolation followed by RT-qPCR.

First, we mechanically isolate intestinal villi and crypts from 8-week-old C57BL/6 mice. New Supplementary Fig. 1a shows that *Znhit1* is mainly expressed in intestinal crypts. Then, we dissociate *Lgr5-EGFP-IRES-creERT2* crypts into single cells and sort *Lgr5*⁺ ISCs (GFP^{hi}), daughter progenitor cells (GFP^{low}) and other crypt cells (GFP^{neg}) using FACS (new Supplementary Fig. 1b). We find that *Lgr5*⁺ ISCs have robust *Znhit1* expression, while their daughter progenitor cells and other crypt cells have significantly reduced *Znhit1* expression (new Supplementary Fig. 1b). This ISC-enriched expression pattern of *Znhit1* supports its primary function in determining the fate of *Lgr5*⁺ ISCs.

Minor issues to be dealt with:

1. Why do the authors not try to knock out *znhit1* in *lgr5* cells directly (using an *lgr5-cre*)? *Olfm4-cre* is a suitable substitute, but still only a substitute.

Response:

Although *Lgr5-EGFP-IRES-creERT2* strain is widely employed to characterize *Lgr5*⁺ ISCs, the expression of *creERT2* is silenced in patches of crypts¹, which makes the strain inappropriate for investigating the contribution of *Lgr5*⁺ ISCs negative regulation to epithelial homeostasis (rapid compensation from adjacent crypts).

To our knowledge, *Olfm4-IRES-eGFPcreERT2* is the best tool to delete gene specifically in *Lgr5*⁺ ISCs throughout the intestinal epithelium with ISCs GFP-marked. Of note, we confirmed key results in *Znhit1*^{fl/fl}; *Olfm4-IRES-eGFPcreERT2* mice, including depletion of *Lgr5*⁺ ISCs (*in vivo*-Fig. 2d,e and *in vitro*-Fig. 2f and Fig. 3f) and changed expression of critical *Znhit1*/H2A.Z target genes (Fig. 3d and new Fig. 3g).

2. An *lgr5* in situ would also be useful to show the extent of its ablation upon *znhit1* ko.

Response:

Thanks for the suggestion. Employing *Lgr5* RNA Scope (quantitative *in situ*), we further confirm that *Znhit1* deficiency leads to ablation of *Lgr5*⁺ ISCs (added in new Fig. 1h).

3. In Figure 1g the genotypes are I think reversed.

Response:

We do appreciate the reviewer for pointing this out. It was a terrible mistake that we reversed the genotype marks when paneling the figure. We correct this in the revised manuscript.

4. In Figure 2 a quantitation of *Lgr5* levels after *znhit1* ablation in *olfm4* cells would be suitable.

Response:

The quantitation of *Lgr5* mRNA level employing RT-qPCR was shown in Fig. 3d. *Znhit1* deletion in *Olfm4*⁺ cells leads to dramatic decrease of *Lgr5* expression in intestinal crypts.

5. Figure 2f should be quantitated, as the effect is not particularly pronounced. Could that be due to escaper crypts that have not recombined *znhit1*?

Response:

In new Fig. 2f, we replace 3-day culture with 7-day culture and provide quantification of organoid buddings along the time for better illustration.

The milder phenotypes of *Znhit1*^{fl/fl}; *Olfm4-IRES-eGFPcreERT2* mice might be due to short-term *Znhit1* knockout (3-day tamoxifen treatment followed by 4-day waiting period, not 4-day tamoxifen treatment followed by 7-day waiting period). We carefully designed this time point to demonstrate that *Znhit1* determines the fate of *Lgr5*⁺ ISCs in a cell autonomous manner. As *Lgr5*⁺ ISCs continuously generate all cell types in intestinal epithelium during homeostasis, longer time point might not exclude the contribution of other cell types to *Lgr5*⁺ ISCs ablation through changing the niche.

6. In figure 3e, what happens to *Lgr5* expression after the treatments? What happens to other *znhit1* ko affected genes, both up- and down-regulated ones?

Response:

We examine the mRNA expression levels of *Lgr5*, *Olfm4*, *Tgfb1* and *Tgfb2* (new Fig. 3g). As shown, *Znhit1* deficiency leads to significant downregulation of *Lgr5* and *Olfm4*, while upregulation of *Tgfb1* and *Tgfb2*. CHIR99021 combined with SB431542 can efficiently rescue the expression of *Lgr5*⁺ ISC signature genes *Lgr5* and *Olfm4*.

7. In figure 4b, there should be a p-value attached, as to how significant the observed overlap is.

Response:

We perform fisher's exact test to evaluate the significance of the overlap between TSS H2A.Z binding genes and *Znhit1*-regulated genes. The statistical test P-value equals 0.05, which is added in Fig. 4b.

8. Are the NGS data deposited somewhere (GEO or other)?

Response:

The raw NGS data were deposited to the NCBI SRA database under accession number SRP148616 (RNA-seq data) and SRP148519 (ChIP-seq data). The data will be released upon publication.

9. The English could do with some polishing. I present just a couple of examples (there are many more, not terribly serious mistakes, in the text, that do need attention):

a. Line 133: which led to following body weight decrease and intestinal epithelium transformation

Should be: which led to concomitant body weight decrease and intestinal epithelium degeneration (or some such)

b. Line 239: play dominate roles

Should be: plays a dominant role (or some such).

Response:

Thanks for the careful reading. We review the manuscript, correct several errors/typos and rephrase some sentences to make the manuscript more readable.

References

1. Schuijers, J., van der Flier, L.G., van Es, J. & Clevers, H. Robust cre-mediated recombination in small intestinal stem cells utilizing the olfm4 locus. *Stem Cell Reports* **3**, 234-241 (2014).
2. Barker, N. Adult intestinal stem cells: critical drivers of epithelial homeostasis and regeneration. *Nat Rev Mol Cell Biol* **15**, 19-33 (2014).
3. Carroll, T.D., Newton, I.P., Chen, Y., Blow, J.J. & Nathke, I. Lgr5(+) intestinal stem cells reside in an unlicensed G1 phase. *J Cell Biol* **217**, 1667-1685 (2018).
4. Munoz, J. *et al.* The Lgr5 intestinal stem cell signature: robust expression of proposed quiescent '+4' cell markers. *EMBO J* **31**, 3079-3091 (2012).
5. Merlos-Suarez, A. *et al.* The intestinal stem cell signature identifies colorectal cancer stem cells and predicts disease relapse. *Cell Stem Cell* **8**, 511-524 (2011).

Reviewers' comments:

Reviewer #1 (Remarks to the Author):

In this revised manuscript assessing the function of *Znhit1* during *Lgr5*⁺ intestinal stem cell (ISC) homeostasis, many new experiments have been performed including some attempts to localize the expression pattern of *Znhit1*, increasing experimental numbers and elaboration of the mechanism underlying *Znhit1*/H2A.Z-mediated transcription. While improved, there are still major issues that in my opinion preclude publication, centered around sufficient demonstration of cell-autonomous *Znhit1* effects within *Lgr5*⁺ ISC.

1. Regarding the mechanism of *Znhit1* effects on the intestinal epithelium, villin-CreER-mediated pan-epithelial *Znhit1* deletion leads to a marked crypt hypertrophy and apparent villus atrophy accompanied by loss of *Lgr5*⁺ ISC (Fig. 1). There is no doubt that the *Znhit1* loss produces *Lgr5*⁺ ISC loss but my question is regarding mechanism. The authors suggest that in the absence of *Lgr5*⁺ ISC there is TA cell expansion. This is possible but if TA cells were truly driving the proliferation in the continued absence of *Lgr5*⁺ ISCs, at extended time points such as d30 in Fig 1d, one might expect eventual crypt loss since the life span of TA cells is only 3-7 days. However, the authors see a massive crypt enlargement at postnatal day 30 in Fig 1d. In Fig. 2, with villin-CreER-mediated *Znhit1* deletion, what happens at time points beyond 11 days – do they die, show more weight loss or lose villi altogether? Please show H&E that shows the entire crypt-villus axis and also Ki67 staining. The authors should discuss alternative possibilities such as the potential role of alternative non-*Lgr5*⁺ ISC populations or even that *Znhit1* is working as a tumor suppressor gene whose deletion is driving crypt enlargement at the expense of differentiation. Since *Znhit1* deletion is being mediated by villin-CreER (i.e. not stem cell specific, but rather pan-epithelial) this hyperproliferation phenotype could be working through cells other than *Lgr5*⁺ ISC.

2. The authors perform a very key experiment in Fig. 3d-g using *Olfm4*-CreER to delete *Znhit1* specifically in *Olfm4*-expressing stem cells, presumably equivalent to *Lgr5*⁺ ISC. This is important to unequivocally demonstrate cell-autonomous function of *Znhit1* in *Lgr5*⁺ ISC and to resolve the nature of the crypt hypertrophy. The deletion of *Znhit1* in *Olfm4*⁺ cells does lead to some reduction in *Lgr5*-eGFP⁺ cells (Fig 3d) and in the rebuttal letter they provide Attached Figure 3 that shows a 10% weight loss at day 11 post tamoxifen, but disappointingly there is no histology provided. In particular, the experiment should have H&E at times beyond day 11 that include the full crypt villus axis and Ki67 staining. Do mice with *Znhit1* deletion in *Olfm4* cells die eventually or show progressive (>10%) weight loss or lose villi altogether? Also it would be helpful to have the Phospho-smad2 staining of *Znhit1*^{fl/fl}; Villin-creERT (fl/fl) in Fig. 3e also be provided for *Znhit1*^{fl/fl}; *Olfm4*-CreER.

3. While it is an improvement that data is presented for *Znhit1* expression in FACS-sorted *Lgr5*-eGFP high versus low cells, consistent with enriched expression in *Lgr5*⁺ ISC, it is still puzzling that there is no attempt at spatial localization in tissue sections ie by in situ hybridization.

4. The Attached Figures in the Rebuttal letter would be very nice additions to the main manuscript or supplemental figures.

Reviewer #3 (Remarks to the Author):

I really appreciate the hard work of the authors. All my concerns were well addressed and I do not have any more critical comment to ask. Therefore, I recommend the publication of the current

manuscript with no delay.

Reviewer #4 (Remarks to the Author):

At this 2nd iteration of the manuscript, the authors have made a generous effort to address some of the concerns we had raised on the first version of the manuscript. While I still agree that the authors convincingly show the involvement of *znhit1* in stem cell maintenance (and, despite their insistence, not specification) in the intestine, some of the issues that we raised have still not been sufficiently addressed: as a result, the manuscript overreaches in its conclusions to highlight a function of *znhit1* specifically in the intestinal stem cells, at the expense of other functions of *znhit* in other cell types. These, in my opinion have not been convincingly excluded by the authors. As such, the manuscript is still not suitable for publication.

My main concerns are the following:

1. While the qPCR data in sup figure 1 do point to an enrichment of *znhit* in the crypts and specifically in *Igr5+* cells, and while I do accept the argument that IHC may not be possible with the antibody they have (although they could have tried other antibodies) it should be possible to produce an in situ picture for *znhit* that would more convincingly show in less manipulated cells whether *znhit* is indeed only expressed in *Igr5+* cells or not (they also have the best controls for this staining in the ko intestines).
 2. There is still not enough clarification on the interplay of *znhit* and *yl1*: I do appreciate that it may be difficult to isolate the phosphorylation site (even though I am still not convinced that it is phosphorylation – it could be ubiquitylation or sumoylation or a combination thereof, the authors do not address that). However, the info that akt inhibition ablates the modification does not in itself add anything to the manuscript: is akt activity or expression deregulated in *znhit* ko mice? If so, in which cells (*Igr5* or others)? In their lists of transcriptionally deregulated genes, are there any clues about signaling pathways or other modifiers that may point to the responsible kinase/modifier impacting *yl1*?
 3. The analysis of *Tgfb* signaling and its involvement in the phenotype is also problematic, in my opinion. After *znhit* ablation, *tgfb* signaling is expanded in the crypt compartment, if I am interpreting the figures correctly. However, if *Igr5* are depleted in the crypt upon *znhit* ko, the *tgfb* staining must come from other cell types; otherwise, we have to assume that *Igr5* cells have just lost *Igr5* marker expression and have transdifferentiated to another cell type.
 4. The issue of specificity is also not adequately addressed: the overlap between h2az binding sites and *znhit1* ko-deregulated genes is decidedly unimpressive. A *znhit 1* ChIP-seq might clarify more robustly where the protein has a true involvement in regulating transcription. While I do understand that I had not raised these points in my first review and would not insist on such experiments, the authors have made only a lackluster effort to address, using suggested computational analyses of the data they have generated, why only a small subset of genes are affected transcriptionally by depletion, why only a small subset of h2az bound genes are affected, and why some genes go up and some down. The histone h3k4 and h3k27 data they present do not go far enough.
- In general, while the manuscript remains interesting, it does not justify its insistence on explaining the effects of *znhit* depletion by stipulating an effect solely on the stem cell compartment. And it does not mechanistically go far enough to explain the effects that to exist at the level of the stem cells. As such it is not ready for publication in a high impact journal.

Figure changes

Revised Figure	Previous Figure	Modification
Fig. 1a		New data
Fig. 1b	Fig. 1a	
Fig. 1c	Fig. 1b,c	Combined
Fig. 2b		New data
Fig. 2c-g	Fig. b-f	
Fig. 2h		New data
Fig. 4e	Fig. 4e	New data added
Fig. 4f	Fig. 4f	New data added
Fig. 5d		New data
Fig. 5e-i	Fig. 5d-h	
Supplementary Fig. 3		New data
Supplementary Fig. 4a,b	Supplementary Fig. 3a,b	
Supplementary Fig. 4c		New data
Supplementary Fig. 5		New data
Supplementary Fig. 6,7	Supplementary Fig. 4,5	
Supplementary Fig. 8		New data
Supplementary Fig. 9-11	Supplementary Fig. 6-8	
Supplementary Fig. 12	Supplementary Fig. 9	New data added

Reviewer #1 (Remarks to the Author):

In this revised manuscript assessing the function of *Znhit1* during Lgr5+ intestinal stem cell (ISC) homeostasis, many new experiments have been performed including some attempts to localize the expression pattern of *Znhit1*, increasing experimental numbers and elaboration of the mechanism underlying *Znhit1*/H2A.Z-mediated transcription. While improved, there are still major issues that in my opinion preclude publication, centered around sufficient demonstration of cell-autonomous *Znhit1* effects within Lgr5+ ISC.

1. Regarding the mechanism of *Znhit1* effects on the intestinal epithelium, villin-CreER-mediated pan-epithelial *Znhit1* deletion leads to a marked crypt hypertrophy and apparent villus atrophy accompanied by loss of Lgr5+ ISC (Fig. 1). There is no doubt that the *Znhit1* loss produces Lgr5+ ISC loss but my question is regarding mechanism. The authors suggest that in the absence of Lgr5+ ISC there is TA cell expansion. This is possible but if TA cells were truly driving the proliferation in the continued absence of Lgr5+ ISCs, at extended time points such as d30 in Fig 1d, one might expect eventual crypt loss since the life span of TA cells is only 3-7 days. However, the authors see a massive crypt enlargement at postnatal day 30 in Fig 1d. In Fig. 2, with villin-CreER-mediated *Znhit1* deletion, what happens at time points beyond 11 days – do they die, show more weight loss or lose villi altogether? Please show H&E that shows the entire crypt-villus axis and also Ki67 staining. The authors should discuss alternative possibilities such as the potential role of alternative non-Lgr5+ ISC populations or even that *Znhit1* is working as a tumor suppressor gene whose deletion is driving crypt enlargement at the expense of differentiation. Since *Znhit1* deletion is being mediated by villin-CreER (i.e. not stem cell specific, but rather pan-epithelial) this hyperproliferation phenotype could be working through cells other than Lgr5+ ISC.

Response:

To address the reviewer's concern that *Znhit1* might work as a tumor suppresser and its deletion might lead to hyperproliferation, we describe the phenotype of *Znhit1*^{fl/fl}; *Villin-creERT* with more details. As shown in new Fig. 2b, inducible knockout of *Znhit1* in intestinal epithelium leads to individual death at day 10-14 post tamoxifen administration (n=9), suggesting Lgr5+ ISC depletion caused by *Znhit1* deletion results in entire epithelium failure. Crypt-villus H&E staining (Supplementary Fig. 4b) and Ki67 staining (new Supplementary Fig. 4c) confirm the crypt enlargement and TA population expansion in these mice before death. These data exactly support the idea that the expanded TA cells cannot maintain intestinal homeostasis due to their short life span.

To explain why some of the *Znhit1*^{fl/fl}; *Villin-cre* mice could survive to postnatal day 30, we examine the expression of *Znhit1* along the postnatal time and find that the survived *Znhit1*^{fl/fl}; *Villin-cre* mice show significant *Znhit1* knockout escape after P18

(the following Attached Fig. 1). This knockout escape is commonly observed in deletion of critical stem cell maintaining factors, including c-Myc¹ and Yap².

Attached Figure 1 | *Znhit1*^{fl/fl}; *Villin-cre* mice show significant *Znhit1* knockout escape after P18. Intestinal crypts were harvested from *Znhit1*^{fl/+}; *Villin-cre* (fl/+) and *Znhit1*^{fl/fl}; *Villin-cre* (fl/fl) mice at indicated time for qRT-PCR to examine the expression of *Znhit1*. Histone H3 was used as an internal control. The statistical data represent mean±s.d. (n=3). *** indicates P<0.001. * indicates P<0.05.

2. The authors perform a very key experiment in Fig. 3d-g using *Olfm4-CreER* to delete *Znhit1* specifically in *Olfm4*-expressing stem cells, presumably equivalent to *Lgr5*⁺ ISC. This is important to unequivocally demonstrate cell-autonomous function of *Znhit1* in *Lgr5*⁺ ISC and to resolve the nature of the crypt hypertrophy. The deletion of *Znhit1* in *Olfm4*⁺ cells does lead to some reduction in *Lgr5-eGFP*⁺ cells (Fig 3d) and in the rebuttal letter they provide Attached Figure 3 that shows a 10% weight loss at day 11 post tamoxifen, but disappointingly there is no histology provided. In particular, the experiment should have H&E at times beyond day 11 that include the full crypt villus axis and Ki67 staining. Do mice with *Znhit1* deletion in *Olfm4* cells die eventually or show progressive (>10%) weight loss or lose villi altogether? Also it would be helpful to have the Phospho-smad2 staining of *Znhit1*^{fl/fl}; *Villin-creERT* (fl/fl) in Fig. 3e also be provided for *Znhit1*^{fl/fl}; *Olfm4-CreER*.

Response:

According to the reviewer's suggestion, we provide the survival curve and detailed histological description of *Znhit1*^{fl/fl}; *Olfm4-IRES-eGFPcreERT2* mice. As shown in the following Attached Fig. 2, *Lgr5*⁺ ISC specific *Znhit1* deletion leads to individual death at day 12-18 post tamoxifen administration (n=7), which is delayed for 2-3 days compared to *Znhit1*^{fl/fl}; *Villin-creERT* mice. Crypt-villus H&E and Ki67 staining at day 11 post tamoxifen administration (before individual death) shows enlarged crypts, expanded TA cells and defective villi (new Supplementary Fig. 5). Besides, phospho-Smad2 staining confirms the TGF-β activation in *Znhit1*^{fl/fl}; *Olfm4-IRES-eGFPcreERT2* crypts (new Supplementary Fig. 6). These data together

reveal that Lgr5+ ISC specific *Znhit1* deletion mimics the phenotype of *Villin-creERT*-mediated entire epithelium deletion, suggesting *Znhit1* mainly functions through supporting Lgr5+ ISCs.

Attached Figure 2 | Lgr5+ ISC specific *Znhit1* deletion leads to individual death. Kaplan–Meier survival curves of *Znhit1*^{+/+}; *Olfm4-IRES-eGFPcreERT2* (+/+) and *Znhit1*^{fl/fl}; *Olfm4-IRES-eGFPcreERT2* (fl/fl) mice following tamoxifen administration (n=7).

3. While it is an improvement that data is presented for *Znhit1* expression in FACS-sorted Lgr5-eGFP high versus low cells, consistent with enriched expression in Lgr5+ ISC, it is still puzzling that there is no attempt at spatial localization in tissue sections ie by in situ hybridization.

Response:

The *Znhit1 in situ* is performed in eight-week-old C57BL/6 mouse intestine section according to the reviewer’s suggestion. As shown in new Fig. 1a, *Znhit1* transcription is restricted to the bottom of intestinal crypts, which is in consistent with our FACS data that Lgr5+ ISCs have greatly enriched *Znhit1* expression (Supplementary Fig. 1). This Lgr5+ ISC-restricted expression pattern of *Znhit1* supports its primary function in determining the fate of Lgr5+ ISCs.

4. The Attached Figures in the Rebuttal letter would be very nice additions to the main manuscript or supplemental figures.

Response:

We appreciate the reviewer’s constructive suggestion. The previous Attached Figures are now integrated into the manuscript as new Fig. 2h (decreased body weight after Lgr5+ ISC specific *Znhit1* deletion) and new Supplementary Fig. 3 (comparable terminal differentiation after *Znhit1* deletion).

Reviewer #3 (Remarks to the Author):

I really appreciate the hard work of the authors. All my concerns were well addressed and I do not have any more critical comment to ask. Therefore, I recommend the publication of the current manuscript with no delay.

Response:

Thanks a lot for the reviewer's appreciation.

Reviewer #4 (Remarks to the Author):

At this 2nd iteration of the manuscript, the authors have made a generous effort to address some of the concerns we had raised on the first version of the manuscript. While I still agree that the authors convincingly show the involvement of *znhit1* in stem cell maintenance (and, despite their insistence, not specification) in the intestine, some of the issues that we raised have still not been sufficiently addressed: as a result, the manuscript overreaches in its conclusions to highlight a function of *znhit1* specifically in the intestinal stem cells, at the expense of other functions of *znhit1* in other cell types. These, in my opinion have not been convincingly excluded by the authors. As such, the manuscript is still not suitable for publication.

My main concerns are the following:

1. While the qPCR data in sup figure 1 do point to an enrichment of *znhit1* in the crypts and specifically in *Lgr5+* cells, and while I do accept the argument that IHC may not be possible with the antibody they have (although they could have tried other antibodies) it should be possible to produce an in situ picture for *znhit1* that would more convincingly show in less manipulated cells whether *znhit1* is indeed only expressed in *Lgr5+* cells or not (they also have the best controls for this staining in the ko intestines).

Response:

The *Znhit1 in situ* is performed in eight-week-old C57BL/6 mouse intestine section according to the reviewer's suggestion. As shown in new Fig. 1a, *Znhit1* transcription is restricted to the bottom of intestinal crypts, which is in consistent with our FACS data that *Lgr5+* ISC's have greatly enriched *Znhit1* expression (Supplementary Fig. 1). This *Lgr5+* ISC-restricted expression pattern of *Znhit1* supports its primary function in determining the fate of *Lgr5+* ISC's.

2. There is still not enough clarification on the interplay of *znhit1* and *yl1*: I do appreciate that it may be difficult to isolate the phosphorylation site (even though I am still not convinced that it is phosphorylation – it could be ubiquitylation or sumoylation or a combination thereof, the authors do not address that). However, the info that akt inhibition ablates the modification does not in itself add anything to the manuscript: is akt activity or expression deregulated in *znhit1* ko mice? If so, in which cells (*Lgr5* or others)? In their lists of transcriptionally deregulated genes, are there any clues about signaling pathways or other modifiers that may point to the responsible kinase/modifier impacting *yl1*?

Response:

In last revision, we identified YL1 as a p-Akt phosphorylation substrate (Fig. 5b) and showed that inhibiting Akt activity ablates YL1 phosphorylation (Fig. 5g). However, as the reviewer pointed out, it was not determined whether *Znhit1* could regulate YL1 phosphorylation through controlling Akt activity. Performing immunoprecipitation, we find that *Znhit1* deletion efficiently abolishes the interaction between p-Akt and YL1 without affecting Akt activity (new Fig. 5d), indicating that *Znhit1* is essential for the binding of p-Akt to YL1 and consequent YL1 phosphorylation.

3. The analysis of Tgfb signaling and its involvement in the phenotype is also problematic, in my opinion. After *znhit* ablation, *tgfb* signaling is expanded in the crypt compartment, if I am interpreting the figures correctly. However, if *lgr5* are depleted in the crypt upon *znhit* ko, the *tgfb* staining must come from other cell types; otherwise, we have to assume that *lgr5* cells have just lost *lgr5* marker expression and have transdifferentiated to another cell type.

Response:

As no significant cell death is observed in intestinal crypts after *Znhit1* deletion (revealed by cleaved caspase 3 staining in new Supplementary Fig. 3a), we propose that *Znhit1*-deficient *Lgr5*⁺ ISCs undergo differentiation but not cell death. Notably, we demonstrate that the differentiated *Lgr5*⁺ ISCs lose the pluripotency as they have decreased expression of multiple stemness markers (*in situ* data in Fig. 1h and RT-qPCR data in Fig. 3b,d) and cannot give rise to organoids in functional assay (Fig. 1f and Fig. 2d,g). These data suggest that the differentiated daughter cells (should be *Ki67*⁺ TA cells as shown in Fig. 1e, new Supplementary Fig. 4c and new Supplementary Fig. 5b) provide the expanded TGF- β signaling.

4. The issue of specificity is also not adequately addressed: the overlap between *h2az* binding sites and *znhit1* ko-deregulated genes is decidedly unimpressive. A *znhit1* ChIP-seq might clarify more robustly where the protein has a true involvement in regulating transcription. While I do understand that I had not raised these points in my first review and would not insist on such experiments, the authors have made only a lackluster effort to address, using suggested computational analyses of the data they have generated, why only a small subset of genes are affected transcriptionally by depletion, why only a small subset of *h2az* bound genes are affected, and why some genes go up and some down. The histone *h3k4* and *h3k27* data they present do not go far enough.

Response:

Thank the reviewer for the constructive comment. We do appreciate the suggestion that *Znhit1* ChIP-seq will benefit the understanding of how *Znhit1* regulates H2A.Z incorporation and target gene transcription. Honestly, we tried several times but the only available *Znhit1* antibody failed to concentrate *Znhit1* protein for IP or ChIP.

To explain why *Znhit1/H2A.Z* deficiency affects the transcription of a small subset of H2A.Z-bound genes and exerts opposite regulatory effects, we expand the H3K4me3 and H3K27me3 analysis to *Mettl3* and *Prmt1*, both of which have TSS H2A.Z enrichment but show no expression alternation after H2A.Z deletion (new Fig. 4e). We find that *Lgr5* TSS region has an original enrichment of H3K4me3 (transcription activation landmark), while *Tgfb1* TSS region has an original enrichment of H3K27me3 (transcription suppression landmark) (Fig. 4f). Both H3K4me3 and H3K27me3 landmarks are efficiently ablated after H2A.Z deletion (Fig. 4f), suggesting that the histone H3 modification status might determine the opposite regulatory effects of H2A.Z on transcription of different genes. Interestingly, as H2A.Z deficiency does not disrupt the balance between H3K4me3 and H3K27me3 on TSS region of *Mettl3* or *Prmt1* (Fig. 4f), the transcription is not affected (Fig. 4e). These data together suggest that H2A.Z specifically controls gene transcription through establishing the competing advantage of either H3K4me3 (for transcription activation) or H3K27me3 (for transcription suppression).

In consistent with our data, recent H2A.Z studies (performed in yeast³, mouse embryonic stem cells⁴ and mouse brain⁵) also showed that most H2A.Z binding genes had no expression change upon H2A.Z removal, indicating the high specificity of transcriptional regulation (either up or down). To our understanding, *Znhit1/H2A.Z* regulates the TSS accessibility through remodeling chromatin, but the potential transcriptional output is determined by various following transcriptional effectors.

In general, while the manuscript remains interesting, it does not justify its insistence on explaining the effects of *znhit* depletion by stipulating an effect solely on the stem cell compartment. And it does not mechanistically go far enough to explain the effects that to exist at the level of the stem cells. As such it is not ready for publication in a high impact journal.

Response:

We appreciate the reviewer's suggestions, which do help us a lot improve the manuscript, especially the mechanisms.

We now provide the *Znhit1 in situ* and FACS data showing the restricted expression of *Znhit1* in *Lgr5+* ISCs. Mechanistically, employing ISC specific *Znhit1* deletion, we show that *Znhit1* recruits p-Akt to phosphorylate YL1 thus maintains the interaction between YL1 and H2A.Z.

We wish we address the reviewer's major concerns appropriately. Thanks!

References

1. Muncan, V. et al. Rapid loss of intestinal crypts upon conditional deletion of the Wnt/Tcf-4 target gene c-Myc. *Mol Cell Biol* 26, 8418-8426 (2006).
2. Cai, J., Maitra, A., Anders, R.A., Taketo, M.M. & Pan, D. beta-Catenin destruction complex-independent regulation of Hippo-YAP signaling by APC in intestinal tumorigenesis. *Genes Dev* 29, 1493-1506 (2015).
3. Yamada, S. et al. The histone variant H2A.Z promotes initiation of meiotic recombination in fission yeast. *Nucleic Acids Res* 46, 609-620 (2018).
4. Hu, G. et al. H2A.Z facilitates access of active and repressive complexes to chromatin in embryonic stem cell self-renewal and differentiation. *Cell Stem Cell* 12, 180-192 (2013).
5. Shen, T. et al. Brain-specific deletion of histone variant H2A.z results in cortical neurogenesis defects and neurodevelopmental disorder. *Nucleic Acids Res* 46, 2290-2307 (2018).

REVIEWERS' COMMENTS:

Reviewer #1 (Remarks to the Author):

The manuscript is much improved. It is reassuring to see the in situ hybridization of *Znhit1* localizing to the crypt bases in Figure 1. It is also helpful to see some of the in vivo phenotypes of the *Olfm4-CreER*-mediated deletion of *Znhit1* with substantial lethality and some degree of histologic analysis.

While this essentially addresses all my concerns, it is still puzzling to me that the authors are not presenting the most dramatic aspects of the *Olfm4-CreER*-mediated phenotype, since this cross is the most supportive of a cell-autonomous role for *Znhit1* in intestinal stem cells. For instance, the *Olfm4-CreER x Znhit1^{f/fl}* survival curve presented as Review Figure 2 shows complete lethality but is for some reason not included in the Main or Supplemental figures. Since there is complete death in this cross by day 18, why not show histology at later time points which would have more severe intestinal histology? The day 7 histology for the *Olfm4-CreER x Znhit1^{f/fl}* cross in Supplemental Figure 5 is not very convincing at all as the complete crypt villus axis is not shown, so replacing these images with better transverse sections would be very helpful. And the day 7 histology for the *Olfm4-CreER x Znhit1^{f/fl}* cross in Supplemental Figure 5 is not nearly as convincing as the Supplemental Figure 4 *Villin-CreER x Znhit1^{f/fl}* cross which has full visualization of the crypt/villus axis and a wider field of view. These simple things would make this paper much more convincing and the authors presumably have this data already.

Reviewer #4 (Remarks to the Author):

The authors have at this stage of the process done enough to alleviate most of my concerns regarding some of the conclusions of this work. As such, provided certain issues are dealt with, I will not stand in the way of its publication.

I will insist on two things: that the authors desist from talking about *znhit* affecting stem cell 'specification' and rather talk about 'maintenance' throughout the text. The results they present do not demonstrate such a role for *znhit*, I am not even sure that the authors understand 'specification' to mean the same thing as I (and others in the field) understand it.

The second point I will insist on is that they should talk about an ISC-'enriched' rather than 'specific' pattern for *znhit* expression. The totality of their data leaves open the probability that *znhit1* is expressed, albeit in lower levels, in other cell types apart from the ISCs and that this extra-ISC expression has some contribution to the phenotypes observed. The authors should leave open that possibility, lest future work puts excessively strong statements made here in an unkind light.

The manuscript would benefit from more polishing of the English, and a bit of scientific editing, as there are phrases that do not convey the intended meaning appropriately. E.g., 'These data together suggest that H2A.Z specifically controls gene transcription through establishing the competing advantage of either H3K4me3 (for transcription activation) or H3K27me3 (for transcription suppression)' is a bit confusing: what advantage? Do the authors mean that H2A.Z allows factors to come in and enhance the depositions of preexisting modifications, either positive or negative? These and other language issues should be dealt with before publication.

Two minor points: Figure 4a is misleading as it is presented: the authors should define distribution of H2A.Z peaks in such a way as to make clear that the vast majority of the peaks is in TSS regions. Either define promoter as -5 kb to +0.5 kb or create a new category called 'TSS' (from -0.5 to +0.5 kb) and redefine 'promoter' as 'promoter-proximal' (-5 kb to -0.5 kb). The classification of many peaks as intronic or exonic, when they are clearly TSS-proximal is misleading. Second, in figure 5d, I am not sure whether some label is missing ('mock' or 'IgG control' and 'IP'). Otherwise, why are there four

lanes instead of two there and why are the +/+ lanes different in terms of pAkt blotting? Is there something I'm missing?

Figure changes

Revised Figure	Previous Figure	Modification
Fig. 2a,h	Fig. 2a,h	Mouse cartoon replaced
Fig. 4a	Fig. 4a	Data reorganized
Fig. 5d	Fig. 5d	Label modified
Supplementary Fig. 5a	Response Figure 2	
Supplementary Fig. 5b	Supplementary Fig. 5	Replaced with new data

Reviewer #1 (Remarks to the Author):

The manuscript is much improved. It is reassuring to see the in situ hybridization of *Znhit1* localizing to the crypt bases in Figure 1. It is also helpful to see some of the in vivo phenotypes of the *Olfm4*-CreER-mediated deletion of *Znhit1* with substantial lethality and some degree of histologic analysis.

While this essentially addresses all my concerns, it is still puzzling to me that the authors are not presenting the most dramatic aspects of the *Olfm4*-CreER-mediated phenotype, since this cross is the most supportive of a cell-autonomous role for *Znhit1* in intestinal stem cells. For instance, the *Olfm4*-CreER x *Znhit1*^{fl/fl} survival curve presented as Review Figure 2 shows complete lethality but is for some reason not included in the Main or Supplemental figures. Since there is complete death in this cross by day 18, why not show histology at later time points which would have more severe intestinal histology? The day 7 histology for the *Olfm4*-CreER x *Znhit1*^{fl/fl} cross in Supplemental Figure 5 is not very convincing at all as the complete crypt villus axis is not shown, so replacing these images with better transverse sections would be very helpful. And the day 7 histology for the *Olfm4*-CreER x *Znhit1*^{fl/fl} cross in Supplemental Figure 5 is not nearly as convincing as the Supplemental Figure 4 Villin-CreER x *Znhit1*^{fl/fl} cross which has full visualization of the crypt/villus axis and a wider field of view. These simple things would make this paper much more convincing and the authors presumably have this data already.

Response:

We do appreciate the reviewer's suggestions to improve the manuscript. Accordingly, we move the survival curve of ISC-specific *Znhit1* deletion from previous Review Fig. 2 to new Supplementary Fig. 5a. Then, we replace Supplementary Fig. 5b with better images (full visualization of the crypt-villus axis and wide view). We wish the more convincing histology could address the reviewer's concern.

Thank the reviewer for the constant help!

Reviewer #4 (Remarks to the Author):

The authors have at this stage of the process done enough to alleviate most of my concerns regarding some of the conclusions of this work. As such, provided certain issues are dealt with, I will not stand in the way of its publication.

I will insist on two things: that the authors desist from talking about znhit affecting stem cell ‘specification’ and rather talk about ‘maintenance’ throughout the text. The results they present do not demonstrate such a role for znhit, I am not even sure that the authors understand ‘specification’ to mean the same thing as I (and others in the field) understand it.

Response:

We appreciate the reviewer’s suggestion. Accordingly, we change “specification” to “maintenance” or “postnatal generation” in the title and throughout the text.

The second point I will insist on is that they should talk about an ISC-‘enriched’ rather than ‘specific’ pattern for znhit expression. The totality of their data leaves open the probability that znhit1 is expressed, albeit in lower levels, in other cell types apart from the ISCs and that this extra-ISC expression has some contribution to the phenotypes observed. The authors should leave open that possibility, lest future work puts excessively strong statements made here in an unkind light.

Response:

Thanks for the insightful comment. Accordingly, we change “ISC-restricted” to “ISC-enriched” in describing the expression pattern of Znhit1 to open the possibility that low expressing Znhit1 might function in other cells.

The manuscript would benefit from more polishing of the English, and a bit of scientific editing, as there are phrases that do not convey the intended meaning appropriately. E.g., ‘These data together suggest that H2A.Z specifically controls gene transcription through establishing the competing advantage of either H3K4me3 (for transcription activation) or H3K27me3 (for transcription suppression)’ is a bit confusing: what advantage? Do the authors mean that H2A.Z allows factors to come in and enhance the depositions of preexisting modifications, either positive or negative? These and other language issues should be dealt with before publication.

Response:

The statement is modified to “These data together suggest that H2A.Z specifically controls gene transcription through permitting regulatory histone H3 methylations” for better understanding.

We examine and polish the English language repeatedly to ensure the clarity and readability.

Two minor points: Figure 4a is misleading as it is presented: the authors should define distribution of H2A.Z peaks in such a way as to make clear that the vast majority of the peaks is in TSS regions. Either define promoter as -5 kb to +0.5 kb or create a new category called 'TSS' (from -0.5 to +0.5 kb) and redefine 'promoter' as 'promoter-proximal' (-5 kb to -0.5 kb). The classification of many peaks as intronic or exonic, when they are clearly TSS-proximal is misleading. Second, in figure 5d, I am not sure whether some label is missing ('mock' or 'IgG control' and 'IP'). Otherwise, why are there four lanes instead of two there and why are the ++ lanes different in terms of pAkt blotting? Is there something I'm missing?

Response:

We do appreciate the reviewer's careful reading. We reorganize the data in Fig. 4a to emphasize the H2A.Z peaks in TSS regions. Previous "promoter", "exon" and "intron" are redefined in figure and text to avoid any misleading. The label of Fig. 5d is indeed missing. We correct this error and perform throughout examination.

Thank the reviewer for the constant help!